# Direction of actin flow dictates integrin LFA-1 orientation during leukocyte migration

Pontus Nordenfelt [1,2,3,4], Travis I. Moore [1,3], Shalin B. Mehta[5,10], Joseph Mathew Kalappurakkal[1,2,6], Vinay Swaminathan[1,2,7], Nobuyasu Koga[8,11], Talley J. Lambert [9], David Baker[8], Jennifer C. Waters[9], Rudolf Oldenbourg [5], Tomomi Tani[5], Satyajit Mayor[1,2,6], Clare M. Waterman[1,2,7] & Timothy A. Springer [1,2,3]

Integrin αβ heterodimer cell surface receptors mediate adhesive interactions that provide traction for cell migration. Here, we test whether the integrin, when engaged to an extracellular ligand and the cytoskeleton, adopts a specific orientation dictated by the direction of actin flow on the surface of migrating cells. We insert GFP into the rigid, ligand-binding head of the integrin, model with Rosetta the orientation of GFP and its transition dipole relative to the integrin head, and measure orientation with fluorescence polarization microscopy. Cytoskeleton and ligand-bound integrins orient in the same direction as retrograde actin flow with their cytoskeleton-binding β-subunits tilted by applied force. The measurements demonstrate that intracellular forces can orient cell surface integrins and support a molecular model of integrin activation by cytoskeletal force. Our results place atomic, Å-scale structures of cell surface receptors in the context of functional and cellular, μm-scale measurements.

[1] Whitman Center, Marine Biological Laboratory, Woods Hole, MA 02543, USA. [2] Physiology Course, Marine Biological Laboratory, Woods Hole, MA 02543, USA. [3] Program in Cellular and Molecular Medicine, Children's Hospital, and Department of Biological Chemistry and Molecular Pharmacology and Medicine, Harvard Medical School, Boston, MA 02115, USA. [4] Division of Infection Medicine, Department of Clinical Sciences Lund, Faculty of Medicine, Lund University, Lund 221 84, Sweden. [5] Eugene Bell Center, Marine Biological Laboratory, Woods Hole, MA 02543, USA. [6] National Center for Biological Sciences, Bangalore 560065, India. [7] Cell Biology and Physiology Center, NHLBI, NIH, Bethesda, MD 20824, USA. [8] Howard Hughes Medical Institute, University of Washington, Seattle, WA 98195, USA. [9] Department of Cell Biology, Harvard Medical School, Boston, MA 02115, USA. [10]Present address: Chan Zuckerberg Biohub, San Francisco, CA 94158, USA. [11]Present address: Institute for Molecular Science, Myodaiji, Okazaki 444-8585, Japan. Pontus Nordenfelt, Travis I. Moore and Shalin B. Mehta contributed equally to this work. Satyajit Mayor, Clare M. Waterman and Timothy A. Springer jointly supervised this work.  Correspondence and requests for materials should be addressed to T.A.S. (email: springer_lab@crystal.harvard.edu)

The integrin lymphocyte function-associated antigen-1 (LFA-1, αLβ2) participates in a wide range of adhesive interactions including antigen recognition, emigration from the vasculature, and migration of leukocytes within tissues[1,2]. Integrin ectodomains assume three global conformational states (Fig. 1a) with the extended-open conformation binding ligand with ~1,000-fold higher affinity than the bent-closed and extended-closed conformations[3–5]. Binding of LFA-1 to intercellular adhesion molecule (ICAM) ligands by the αI domain in the integrin head is communicated through the β-subunit leg, transmembrane, and cytoplasmic domains to the actin cytoskeleton via adaptors such as talins and kindlins that bind specific sites in the β-subunit cytoplasmic domain[6]. As reviewed[7,8], measurements of traction force on substrates and more specific measurements of force within ligands and cytoskeletal components have suggested that integrins transmit force between extracellular ligands and the actin cytoskeleton. Forces on the cytoplasmic domain of the LFA-1 β2-subunit have been measured in the 1–6 pN range and associated with binding to ligand and the cytoskeleton[9].

Tensile force exerted through integrins has the potential to straighten the domains in the force-bearing pathway and align them in the direction of force exertion. A strong candidate for the source of this force is actin retrograde flow, which is generated through actin filament extension along the membrane at the cell front[10]. If observed, such alignment would help discriminate among alternative models of integrin activation. Some models

suggest that binding of the cytoskeletal adaptor protein talin to the integrin β-subunit cytoplasmic domain is fully sufficient to activate high affinity of the extracellular domain for ligand[11,12]. Other models, supported by steered molecular dynamics (SMD) and measurements in migrating cells, have proposed that tensile force stabilizes the high-affinity, extended-open integrin conformation because of its increased length along the tensile force-bearing direction compared to the other two integrin conformations (Fig. 1a)[3,9,13–15]. Recently, measurements of the intrinsic affinity and free energies of the three conformational states of integrin α5β1[5] were used to thermodynamically demonstrate that tensile force is required to provide ultrasensitive regulation of integrin adhesiveness[16]. The thermodynamic calculations show that inherent in the three conformational states of integrins is a mechanism by which integrin adhesiveness can be activated when the integrin simultaneously binds the actin cytoskeleton and an extracellular ligand that can resist cytoskeleton-applied force. Thus, the same intracellular effectors that regulate actin dynamics can simultaneously and coordinately regulate cell adhesion to provide the traction for cellular chemotaxis and migration. Furthermore, directional migration is a critical aspect of immune cell function, and alignment of integrins by activation would provide a mechanism for directional sensing. The traction force model of integrin activation is well supported by structural biology and thermodynamic measurements of integrins both in solution and on the surface of intact cells[16]. In this paper, we have tested the traction force model using an

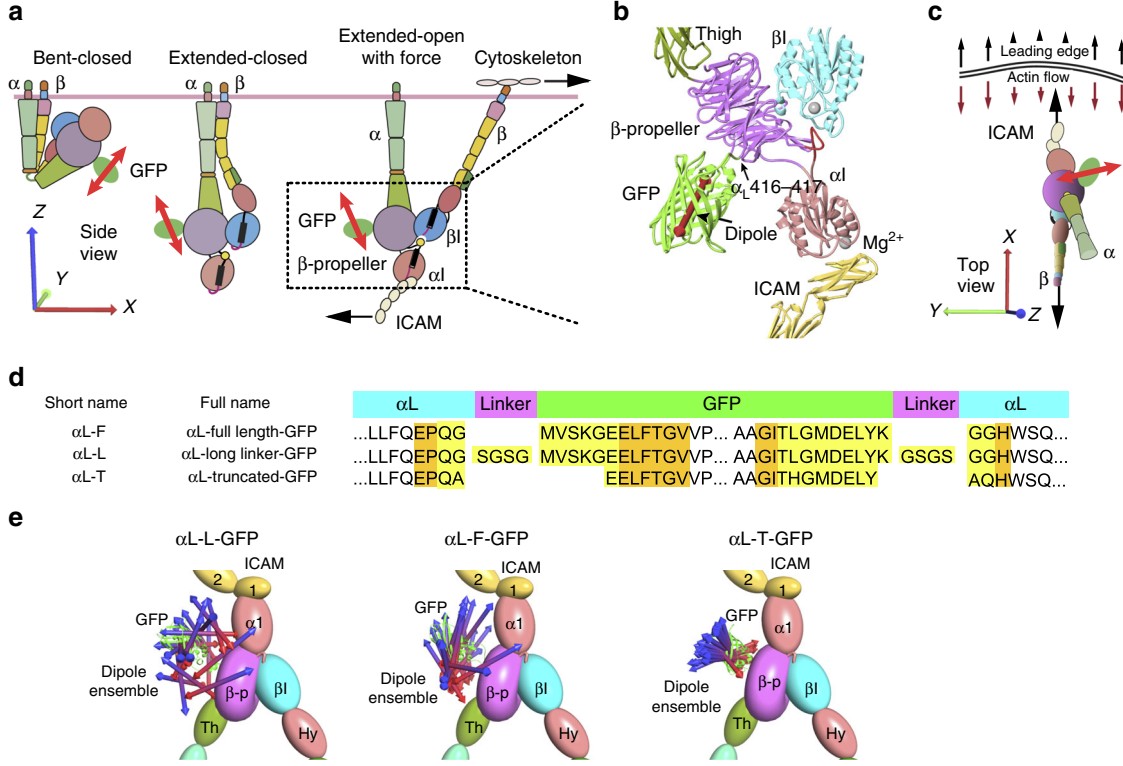

**Fig. 1** Integrins, GFP fusions, and modeling GFP and transition dipole orientation with Rosetta. **a** Three global conformational states of integrins[2]. Cartoons depict each integrin domain and GFP with its transition dipole (red double-headed arrows). **b** Ribbon diagram of the integrin headpiece of αL-T bound to ICAM-1. The GFP insertion site in the β-propeller domain is arrowed. Dipole is shown in red. **c** Cartoon as in **a** of ICAM-engaged, extended-open LFA-1 showing direction of leading edge motion and actin flow. Large arrows show pull on integrin-β by actin and resistance by ICAM-1. Axes shown in **a**, **c** are similar to those in the reference state in Fig. 6. **d** Sequences and boundaries used in GFP-LFA-1 fusions. Highlighted residues were completely modeled by Rosetta to link GFP to the integrin (yellow) or altered in sidechain orientation only to minimize energy (orange). **e** Orientation of the transition dipole in GFP-LFA-1 fusions. Integrin domains are shown as ellipsoids or torus and GFP is shown in cartoon for 1 ensemble member. GFP transition dipoles are shown as cylinders with cones at each end for 20 representative Rosetta ensemble members, with the asymmetry of GFP referenced by using different colors for the ends of transition dipoles (which themselves have dyad symmetry)

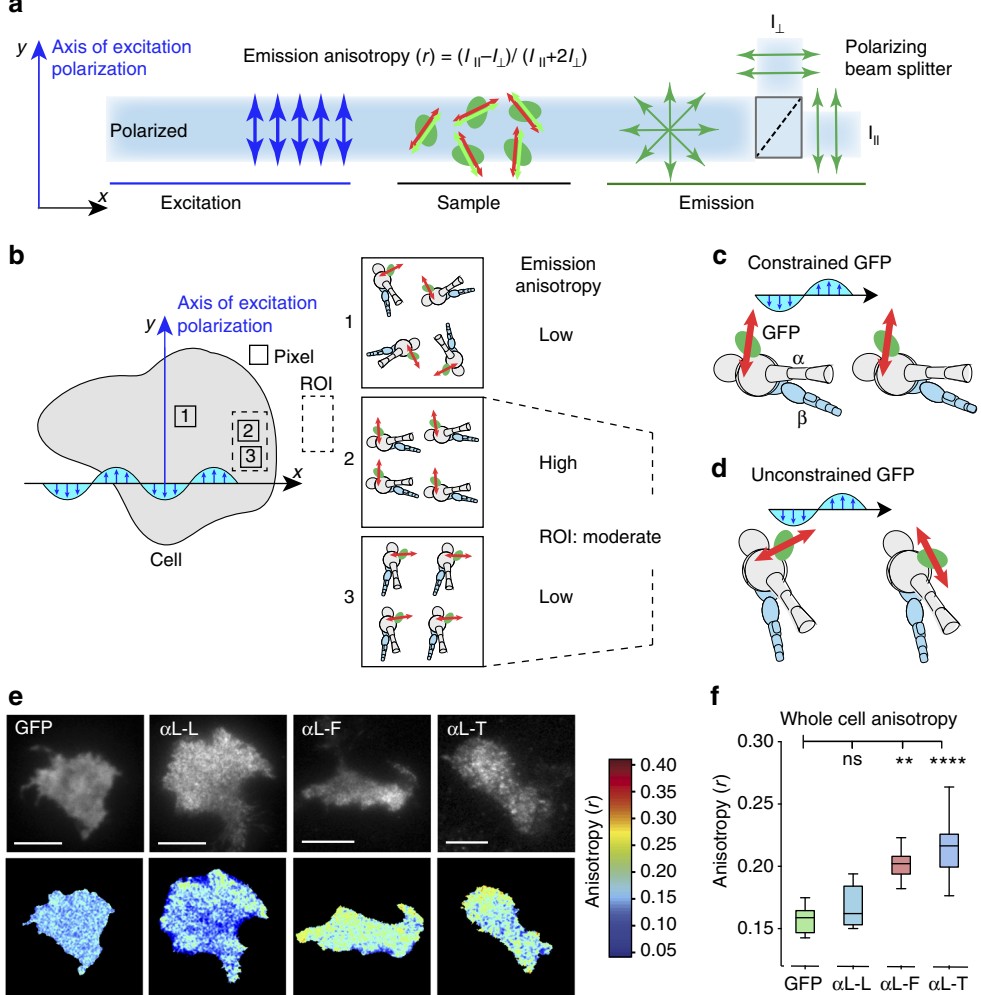

**Fig. 2** Emission anisotropy of GFP-LFA-1 fusions. **a** Schematic of emission anisotropy TIRF microscopy[27]. The transition dipole of GFP has an excitation dipole (green) that is very close in orientation to its emission dipole (red)[28,29]. $I_{||}$ and $I_{\perp}$ are emission intensities parallel and perpendicular to the direction of polarized excitation. **b**–**d** Schematics showing a cell and integrins in same microscope xy plane as in **a** and different outcomes in emission anisotropy. The excitation (electric) field of polarized light is shown as a blue wave. **b** Outcomes using a constrained integrin-GFP fusion. Depending on the orientation of integrins within pixels (1–3) or regions of interest (ROI), the emission anisotropy will change as indicated. **c**, **d** Outcomes with aligned integrins with constrained **c** or unconstrained GFP **d**. **e** Representative images from movies of Jurkat T cells stably expressing GFP-LFA-1 fusions migrating on ICAM-1. Each pair of panels (scale bars are 5 μm) shows total GFP fluorescence intensity (upper) and anisotropy (lower, color scale to right). **f** Emission anisotropy of GFP-LFA-1 fusions, averaged over Jurkat T cells migrating in random directions, in at least five independent experiments. Box plots show the full range (whiskers) of observations with median as line and 25–75 percentile range boxed. Kruskal–Wallis test with multiple comparison correction gave the indicated P values. N (number of cells) from left was 18, 22, 17, 37. *$p < 0.05$; **$p < 0.01$; ****$p < 0.0001$

orthogonal technique that is highly functionally relevant: fluorescent imaging of integrins as they provide the traction for cell migration in living cells.

Here, we test a key prediction of the cytoskeletal force model of integrin activation: that the tensile force exerted through integrins between the actin cytoskeleton and extracellular ligands as they function in cell migration causes them to assume a specific orientation and tilt on the cell surface relative to the direction of pulling on the integrin by actin retrograde flow. Actin flow is known to be locally aligned for migrating fibroblasts and epithelial cells[17,18]. Actin is also known to flow centripetally in immune synapses formed by lymphocytes; furthermore, integrin LFA-1 also moves centripetally but at a lower speed than the actin, as inferred from the movement of its ligand ICAM-1 in planar bilayers on the other side of the immune synapse[19]. Our measurements in this manuscript on integrin orientation on cell surfaces also provide an opportunity to correlate crystal structures of integrins at the Å length scale with microscopic measurements

on integrin-bearing cells at the micron length scale. Integrating measurements at such different length scales is a long-standing goal of many fields of biological research. Although integrins like other membrane proteins are generally free to rotate in the plane of the membrane, tensile force would cause an integrin to orient in the same direction as the pulling force. Like most membrane proteins, integrins are drawn in cartoons (as in Fig. 1a) as projecting with their leg-like domains normal to the plasma membrane; however, resting integrins are free to tilt[20] and force could tilt the integrin far from the membrane normal. In general, despite a wealth of structures for membrane protein ectodomains, little is known about ectodomain orientation on cell surfaces.

In this work, we make use of previous structural studies on integrins[2,4,21,22], and orient these structures in a reference frame that corresponds to the plasma membrane of a migrating lymphocyte. In addition to general structural knowledge on many integrin families[2,4], we make specific use of crystal structures for the αI domain of LFA-1 bound to ICAMs, the LFA-1 headpiece,

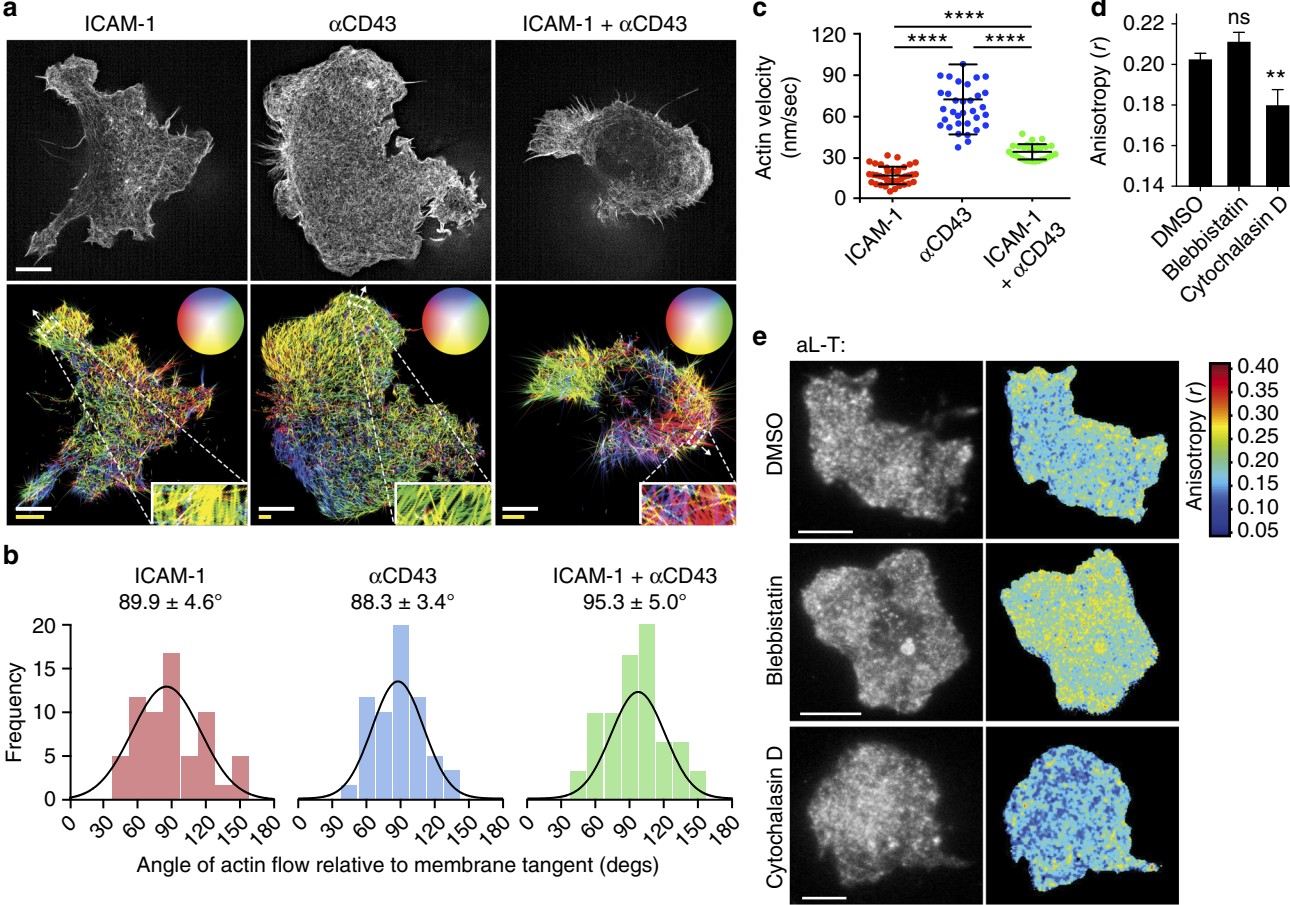

**Fig. 3** Actin flow dynamics, orientation, and relation to GFP-LFA-1 anisotropy in migrating T cells. **a** Representative frames of structured illumination microscopy movies (above) of the actin cytoskeleton visualized with lifeact-mNeonGreen in Jurkat T cells migrating on ICAM-1 (10 µg/ml), anti-CD43 IgG (10 µg/ml), or their mixture (10 µg/ml each). Optical actin flow vector maps of the same cells are shown below with zoom insets of representative areas. Vectors encode flow direction by color (circular keys in the direction from center of circle to perimeter) and velocity by length. Scale bars encode dimensions in the micrograph (white, 5 µm) and velocity (yellow, 30 nm/s). **b** Actin flow direction relative to tangent of leading edge membrane. Bins of 15° are shown with Gaussian fit and mean ± SEM. ICAM-1, $N = 62$; αCD43, $N = 63$; ICAM-1 + αCD43, $N = 67$. ROI ($N$) came from six to eight cells. **c** Leading edge actin flow velocity from optical flow analysis. Plots show data on each cell. Bars show mean ± s.d. Two-tailed Mann–Whitney tests all show $p < 0.0001$ (****). ICAM-1, $N = 39$; αCD43 $N = 38$; and ICAM-1 + αCD43, $N = 25$. **d, e** Jurkat T cells migrating on ICAM-1 (10 µg/ml) expressing αL-T-GFP were treated with DMSO as control, blebbistatin (100 µM), or cytochalasin D (100 nM) and fixed. **d** Whole-cell emission anisotropy analyzed as in Fig. 2e with error bars showing SEM of three independent experiments. Mann–Whitney test for DMSO ($N = 37$) vs. blebbistatin ($N = 9$) was 0.130 and for DMSO vs. cytochalasin D ($N = 14$) was 0.003. **$p < 0.01$. **e** Representative total fluorescence intensity ($I_\parallel + 2I_\perp$, left) and anisotropy (right). Scale bars: 5 µm

and two states of the bent ectodomain of the LFA-1 (αLβ2) relative, αXβ2[2,21,22]. We also use negative stain EM class averages showing the bent-closed, extended-closed, and extended-open conformations of the αLβ2 and αXβ2 ectodomains[2]. These structures together with those of green fluorescent protein (GFP) have guided development here of constrained integrin-GFP fusions and prediction of the orientation of GFP and its fluorescent excitation/emission transition dipole relative to the integrin using Rosetta[23]. Two different types of fluorescent microscopes provide similar measurements of the orientation of the transition dipole relative to the direction of actin flow. Integrin-ligand engagement in combination with cytoskeletal force results in spatially ordered organization of LFA-1 in the protrusive lamellipodial region and is dependent on the movement vector of the underlying actin cytoskeletal framework[19,24,25]. The results show that actin flow from the leading edge dictates a specific molecular orientation on the cell surface of LFA-1 and support the cytoskeletal force model of integrin activation.

## Results

### Design and simulation and testing of constrained LFA-1-GFP.

To report integrin orientation on cell surfaces, we inserted GFP into a loop of the integrin β-propeller domain (Fig. 1a–d). This allows monitoring of the orientation of both the β-propeller and βI domains, which come together over a large, highly stable, rigid interface to form the integrin head. The β-propeller was chosen because of its rigid structure, its lack of participation in integrin conformational change[2], and the availability of a previously validated insertion position that is remote from other integrin domains in all three conformational states[26]. We tested multiple fusions, including αL-L in which residues were added to increase flexibility, and those that deleted residues from N- and C-terminal segments of GFP that are disordered or vary in position among GFP crystal structures and were designed to constrain GFP orientation, including αL-T (Fig. 1d, e and Supplementary Table 1). We modeled with Rosetta any introduced linker residues, residues that vary in position in independent GFP structures, and residues in LFA-1 adjacent to the inserted GFP (Fig. 1d). A wide range of possible

orientations of the two connections between LFA-1 and GFP was effectively sampled using polypeptide segments from the protein databank, selecting those that enabled connections at both the N and C termini of GFP to be closed, and then minimizing the energy of the system with respect to the degrees of freedom of the connecting linkers[23]. The distribution of dipole orientations in the resulting ensembles (Fig. 1e) provides a range of orientations in which the actual orientation should be included, and may approximate the contribution of variation in GFP-integrin orientation to a decrease in emission anisotropy measurable with fluorescence polarization microscopy.

We used emission anisotropy total internal reflection fluorescence microscopy (EA-TIRFM)[27] (Fig. 2a) to examine GFP-LFA-1 fusions stably transfected in Jurkat T lymphocytes that were allowed to randomly migrate on ICAM-1 substrates in the xy focal plane of the microscope (Fig. 2b). In EA-TIRFM, s-polarized light excites fluorophores with dipoles that are parallel to the electric field of incoming light, and the parallel and perpendicular components of the emission fluorescence are recorded on separate cameras (Fig. 2a). The excitation and emission dipoles of GFP are highly aligned to one another and may be referred to collectively as the transition dipole[28,29]. Furthermore, the difference in time between excitation and emission is short relative to tumbling time for GFP[27] (and integrin-GFP fusions). The high NA objectives used in TIRF somewhat lower the absolute values of anisotropy[27] (Supplementary Fig. 1); therefore, we focus here on comparisons of the relative values of anisotropy. Our measurements of fluorescence intensity are for the ensemble of LFA-1-GFP fusions within individual pixels (Fig. 2b). For constrained GFP-LFA-1 fusions that are aligned with one another and when they are aligned in a direction such that the transition dipole is parallel to the electric field of excitation, more emitted fluorescence will be recorded on the parallel than perpendicular camera (Fig. 2c), resulting in higher emission anisotropy than when the dipoles are more normal to excitation (Fig. 2b, compare pixels 2 and 3). In contrast, variation in GFP orientation relative to the integrin in unconstrained GFP (Fig. 2d) or random orientation of constrained GFP-LFA-1 fusions in the plane of the membrane (Fig. 2b, pixel 1) will lead to similar intensities recorded in the parallel and perpendicular cameras, and hence lower emission anisotropy.

To test the Rosetta modeling predictions of GFP constraint, three GFP-integrin fusions were studied on the surface of live Jurkat T lymphoblasts migrating on ICAM-1 substrates (Fig. 2e). Emission anisotropy was highest for the truncated construct, αL-T; intermediate for the full length construct, αL-F; and lowest for the construct with added linker residues, αL-L, which had an emission anisotropy comparable to cytoplasmic GFP (Fig. 2e, f). These findings were in agreement with Rosetta results, which showed that αL-T ensemble members had well aligned transition dipoles (Fig. 1e) and the highest calculated polarization factor (Supplementary Table 2). Anisotropies of aL-F and aL-T-GFP that are substantially higher than for GFP imply that the integrins are not oriented randomly about an axis normal to the plane of membrane, but are aligned with one another in individual pixels (Fig. 2b, pixels 2 and 3); however, integrin orientation might nonetheless differ between pixels (Fig. 2b, ROI). The lack of alignment seen with αL-L compared to the increasing alignment seen with αL-F and αL-T suggests that the orientation of the GFP transition dipole in αL-L is not correlated with the orientation of the integrin (Fig. 2d) in agreement with the lack of orientation seen in αL-L Rosetta ensembles (Fig. 1e). At high fluorophore concentrations, anisotropy can be reduced by homo-Förster resonance energy transfer[27]. However, LFA-1-GFP anisotropy was independent of fluorescence intensity, showing little or no homo-Förster resonance energy transfer (Supplementary Fig. 2).

**Quantitative description of actin flow in migrating T cells.** We hypothesized that force application by the cytoskeleton would orient integrins; therefore, we measured actin flow in migrating lymphocytes prior to determining LFA-1-GFP orientation. Actin flow is known to be retrograde and normal to the leading edge in migrating fibroblasts and epithelial cells[17,18,30]. Much less is known about actin flow in migrating lymphocytes, although it is known to be retrograde within the immunological synapse in non-migrating T cells[19,31]. We measured actin dynamics in migrating T cells expressing lifeact-mNeonGreen using the super-resolution capabilities of structured illumination microscopy (SIM). Actin flow velocity and direction were determined using optical flow analysis[32,33] of the texture maps generated by time-lapse SIM along the leading edge relative to the membrane. Migrating T cells lacked organized actin stress fibers characteristic of many other cells, such as epithelial cells and fibroblasts (Fig. 3a). Optical flow analysis of time-lapse movies (Fig. 3a) showed that flow was retrograde at close to 90° relative to the leading edge (Fig. 3b) with a velocity of 17 ± 1 nm/s (mean ± s.d.) on ICAM-1 substrates (Fig. 3c, Movie 1) as confirmed and consistent with kymographic analysis[9,34] (Supplementary Fig. 3). On the non-integrin substrate, anti-CD43, actin flow was significantly faster (71 ± 4 nm/sec, Movie 2) compared with ICAM-1, whereas flow on mixed ICAM-1 and anti-CD43 substrates was intermediate in velocity (34 ± 2 nm/sec) (Fig. 3c, Movie 3). We do not believe that the large apparent spread in actin flow orientation around the mean value of 90° in Fig. 3b is meaningful for several reasons. Measuring actin flow is always challenging; the alternative technique of speckle microscopy utilizes extensive denoising and averaging[35]; much less denoising is applied in optical flow analysis[33]. Furthermore, assigning the membrane tangent was often challenging as exemplified in the highly irregularly shaped cell on ICAM-1 in Fig. 3a. Finally, visual inspection confirmed the 90° value. In Fig. 3a, actin flow direction in the cells is color coded; direction from the center of the inset circles is coded as the color at the perimeter of the circles. Thus, despite irregularity in shape of the cell on ICAM-1 in Fig. 3a, in the orientation shown in the figure, flow at the top of cell is largely towards the bottom (yellow), flow on the right of the cell is largely toward the left (red), and flow on the left of the cell is largely toward the right (green). Actin flow in migrating lymphocytes is thus centripetal, as previously found in lymphocyte immune synapses[19,31].

The ability of ICAM-1 on substrates to slow retrograde actin flow, and the intermediate results with mixed anti-CD43 and ICAM-1 substrates, strongly suggest that LFA-1 is mechanically linked to retrogradely flowing actin. Activating LFA-1 with $Mn^{2+}$ increased actin flow velocity (28 ± 2 nm sec$^{-1}$) relative to untreated cells (Supplementary Fig. 3), suggesting that artificial activation with $Mn^{2+}$ bypasses the need for integrin association with actin for integrin activation, and leads to a decrease in LFA-1 association with the cytoskeleton. The finding here that ICAM-1 immobilized on a glass coverslip can link through LFA-1 to the actin cytoskeleton is in agreement with previous studies on immunological synapses formed on bilayers in which ICAM-1 can diffuse and be dragged by LFA-1 in a centripetal direction dictated by actin flow[19,31]. However, actin flow in lymphocytes on lipid bilayers is more rapid at ~ 100–300 nm s$^{-1}$, and the ICAM-1 is dragged along at ~40% of this rate, demonstrating a clutch-like connection[36]. Although indirect regulatory mechanisms cannot be ruled out, the most straightforward interpretation of our results is that flowing actin is slowed by the linkage of LFA-1 to both the actin cytoskeleton inside the cell and ICAM-1 on the substrate.

We further tested the role of the actin cytoskeleton in integrin alignment by disrupting two prominent drivers of actin flow,

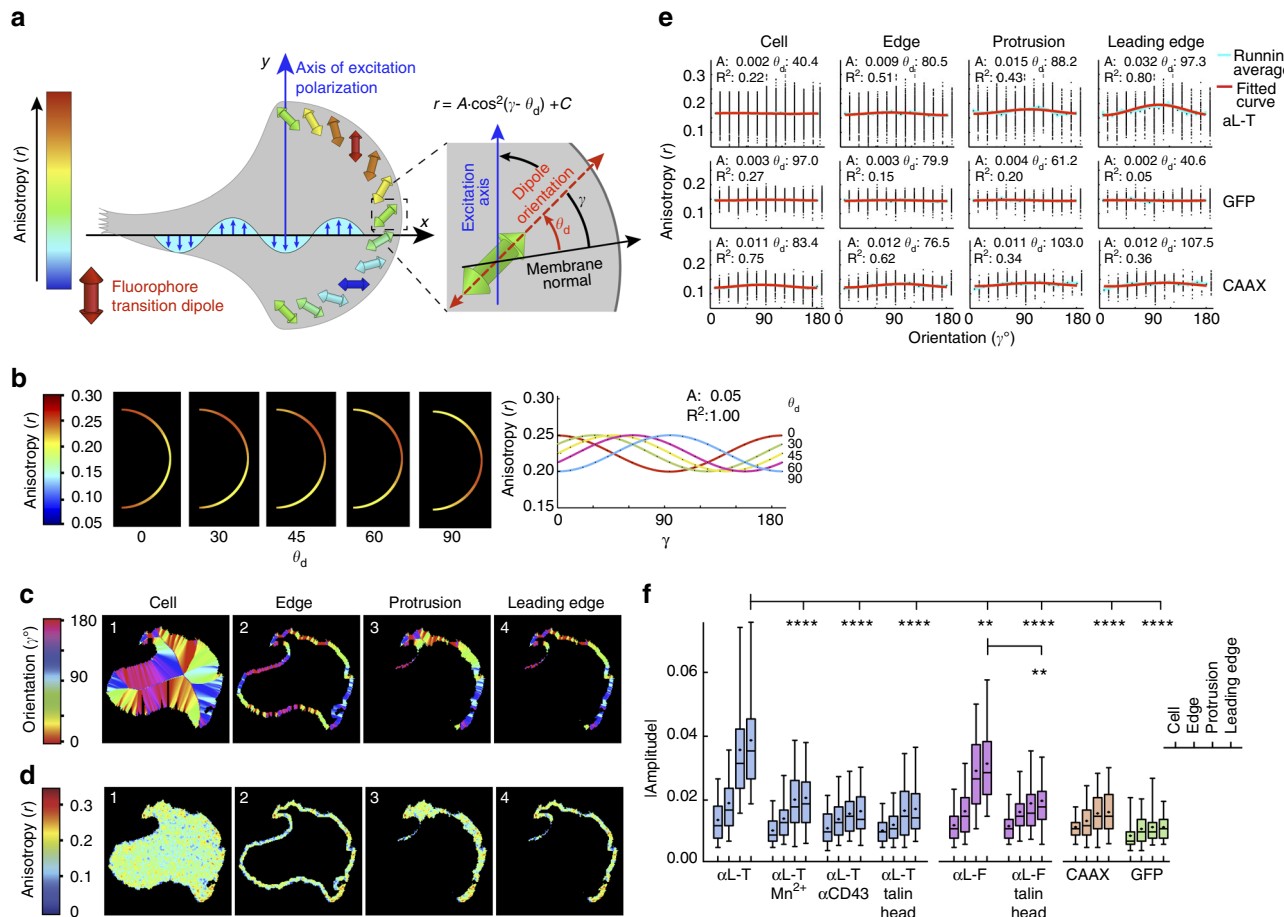

**Fig. 4** Orientation of LFA-1 in the leading edge of migrating cells by EA-TIRFM. **a** Schematic showing relation between excitation polarization orientation, transition dipole orientation, and emission anisotropy in EA-TIRFM. Their quantitative relationship is described by the $\cos^2$ function (inset equation) where $r$ is anisotropy, $A$ is the absolute amplitude of angular dependence and reports the degree of GFP dipole alignment; $\gamma$ is the angle between the membrane normal and the excitation axis; $\theta_d$ is the angle between the membrane normal and transition dipole and is equivalent to the phase shift in the cosine function; and C is isotropic background. **b** Simulation of ability to find a predefined dipole orientation angle relative to membrane normal in an idealized leading edge. Test data were generated with a fixed amplitude (0.05) and five $\theta_d$ angles. Left: simulated leading edge images with anisotropy color-coded according to the key. Right: fitted curves. **c, d** Representative segmentation (cell, edge, protrusion, and leading edge) of a migrating Jurkat T cell expressing $\alpha$L-T-GFP with corresponding maps of orientation relative to the cell midline **c** and anisotropy **d** in the segmented regions (panels 1–4). Scale bar = 5 μm. **e** Emission anisotropy of cell, edge, protrusion, and leading edge segments for $\alpha$L-T, cytosolic GFP, and GFP-CAAX (Supplementary Figs. 5–8) fit to the $\cos^2$ function. Plots show anisotropy vs. orientation relative to the cell midline for each pixel in individual, representative cells (from **c** and **d** for $\alpha$L-T), the running average (blue) and fit (red). Values for the fit parameters ($A$ = absolute amplitude, $\theta_d$ = phase shift, $R^2$ = goodness-of-fit) are displayed in each plot. **f** Absolute amplitudes of emission anisotropy fits to the $\cos^2$ relationship. Box plots show the full range (whiskers) of observations with median as line and 5–95 percentile range boxed. Kruskal–Wallis test with multiple comparison correction to $\alpha$L-T leading edge data ($N = 206$) gave $P$ values from left to right: < 0.0001 ($N = 85$); < 0.0001 ($N = 52$); < 0.0001 ($N = 58$); 0.0033 ($N = 185$); < 0.0001 ($N = 71$); < 0.0001 ($N = 83$); < 0.0001 ($N = 55$) with $N$ leading edges. See Supplementary Table 3 for more details. A two-tailed Mann–Whitney test of $\alpha$L-F in absence and presence of talin head gave a $P$ value of 0.0059. **$p < 0.01$, ***$p < 0.0001$

contractility, and polymerization. Blebbistatin, an inhibitor of myosin-dependent actin filament contractility, had no effect on $\alpha$L-T anisotropy (Fig. 3d, e), consistent with the lack in T cells of actin stress fibers. In contrast, cytochalasin D, an inhibitor of actin polymerization, significantly decreased LFA-1 anisotropy (Fig. 3d, e). Together, these results show that an intact cytoskeleton is required for LFA-1 anisotropy and suggest that actin polymerization, a mechanism operative in the lamellipodium to generate retrograde flow that is independent of actomyosin contraction[17], is important for LFA-1 anisotropy and hence alignment.

**LFA-1 is oriented relative to actin retrograde flow**. Having shown that $\alpha$L-T and $\alpha$L-F LFA-1-GFP fusions were aligned within individual pixels as measured by fluorescence anisotropy,

we next determined whether integrins in larger regions of interest (ROI) near cell leading edges were aligned with one another (ROI, Fig. 2b), and measured the "orientation" of their transition dipoles relative to the leading edge and hence relative to the retrograde direction of actin flow. To do this we employed two independent microscopy techniques.

We first used the photoselective properties of EA-TIRFM[27] to measure orientation. Fluorescence anisotropy shows $\cos^2$ dependence on the angle between the electric field of the polarized light and the fluorophore transition dipole[37,38]. To validate our microscopy system, we first imaged actin filaments, either assembled in vitro and stained with SiR-actin, or stained with Alexa488-phalloidin in migrating T cells (Supplementary Fig. 4). As expected, the value of anisotropy ($r$) was dependent on the angle of the filaments relative to the axis of excitation polarization

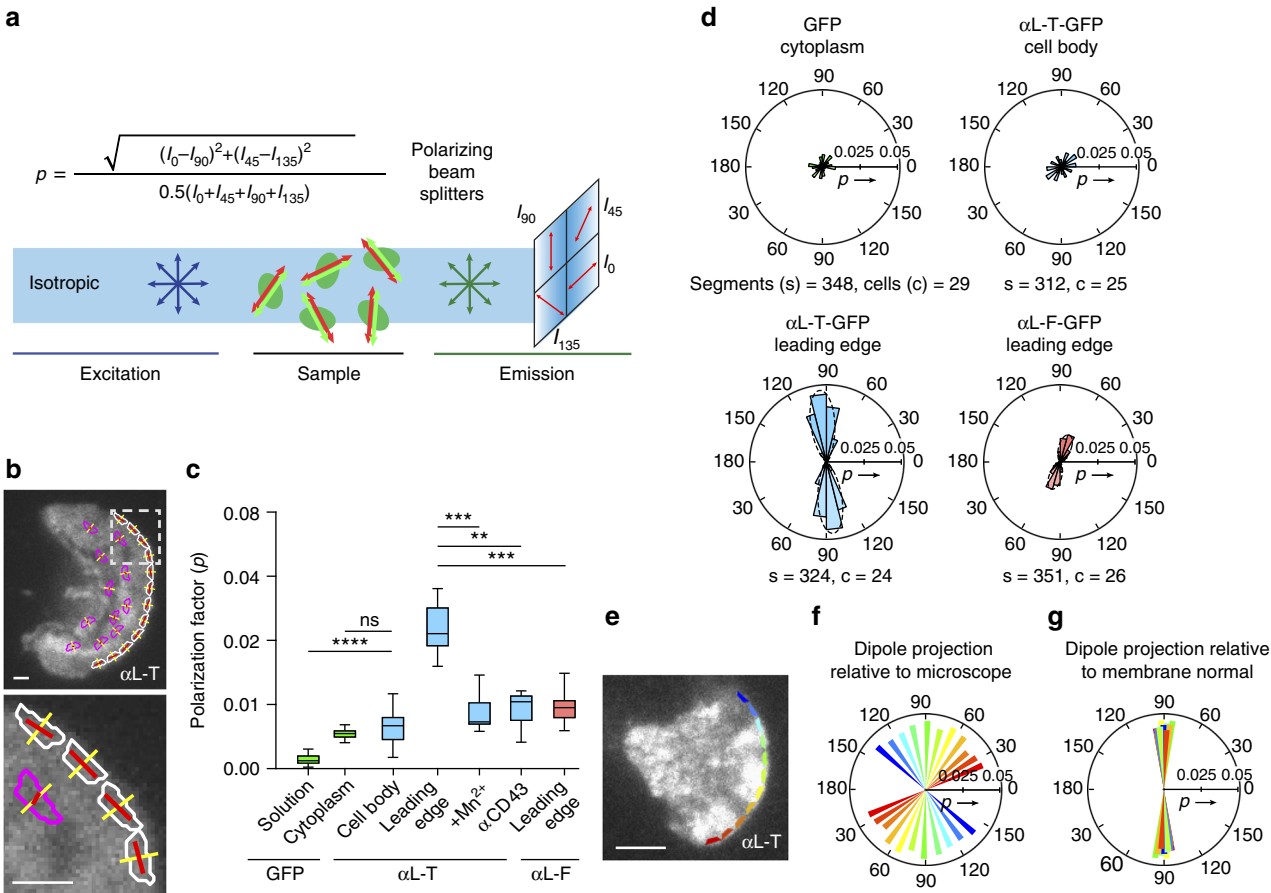

**Fig. 5** Integrin alignment and orientation in migrating T cells measured by instantaneous FluoPolScope. **a** Schematic of Instantaneous FluoPolScope[40] and equation for polarization factor $p$. **b** Representative total fluorescence intensity image of αL-T Jurkat T cell migrating on ICAM-1 (10 μg/ml) with overlay of ROIs (white = leading edge, magenta = cell body), normal to leading edge tangent (yellow), and average GFP emission dipole orientation with length proportional to polarization factor (red). Scale bars = 1 μm. Panel below is enlarged from dashed area. **c** Polarization factors of GFP and GFP-LFA-1 fusions in T cells migrating in random directions. Box plots show the full range (whiskers) of observations with median as line and 25–75 percentile range boxed. Kruskal–Wallis test with multiple comparison correction of leading edges gave the indicated $P$ values to αL-T using $N$ number of cells shown in **d**. **$**p <$ 0.01, **$***p <$ 0.001, $****p <$ 0.0001. **d** Radial histograms of GFP transition dipole projection in image plane relative to the membrane normal ($\theta_d$). Each radial histogram wedge shows mean intensity-weighted polarization factor ($p$) in 15° bins (solid outlines) and is reflected to represent the dyad symmetry axis of the dipole. Dashed propeller-shaped outlines in the lower two panels show the fit of the data to a circular Gaussian. **e–g**, **e** A representative αL-T Jurkat cell migrating on ICAM-1 was segmented as shown in **b**. Segments are color coded in blue-red rainbow around the leading edge and are shown in same color in **f**, **g**. Each segment is represented as a reflected wedge at the angle of its experimentally measured emission dipole projection with respect to the $x$ axis **f** or with respect to the membrane normal ($\theta_d$) **g**

(Supplementary Fig. 4b, c). The value of anisotropy ($r$) was fit to the inset equation in Supplementary Fig. 4a with the angles $\gamma$ and $\theta_d$ defined relative to the long axis of actin filaments. Fits of the $\cos^2$ dependence of anisotropy gave $\theta_d = 86°$ ($R^2 = 0.96$) for SiR-actin and $-2.9°$ ($R^2 = 0.91$) for Alexa488-phalloidin (Supplementary Fig. 4b, c). Because actin filaments are helical, bound fluorophores have cylindrical symmetry, which gives $\theta_d$ values of either 90° or 0°. Our results agree with measured $\theta_d$ values of 0° for Alexa488-phalloidin[39,40] and suggest a $\theta_d$ value of 90° for SiR-actin, as further validated below. These results with actin filament fluorescence polarization validated the use of EA-TIRFM to determine the orientation of LFA-1 in migrating T cells.

We next measured the angular $\cos^2$ dependence of LFA-1-GFP anisotropy using EA-TIRFM to test the hypothesis that integrin engagement to an immobilized ligand and the cytoskeleton would cause the integrin and its associated GFP transition dipole to adopt a specific orientation relative to the direction of actin retrograde flow. Having already established that flow is normal to the leading edge of migrating cells, we tested the angular $\cos^2$ dependence of fluorescence anisotropy on the orientation of the

leading edge relative to polarized excitation (Fig. 4a). We validated our analysis pipeline by generating ideal images of leading edge protrusions with four distinct dipole orientations relative to the leading edge and found the correct dipole angle in each case (Fig. 4b). Migrating cells in movies were segmented into whole-cell, edge, protruding, and leading edge (lamellipodium) regions (Fig. 4c, d, steps 1–4). Geometric shape-based orientation relative to the cell edge was determined by the angle from the cell edge to mid-region (Fig. 4c, see Methods), making it possible to test for angular dependence of anisotropy within each cell. Fits to the $\cos^2$ function for αL-T showed that the extent of angular dependence, i.e., amplitude (A), increased towards the leading edge (Fig. 4e, see also Supplementary Figs. 5–8, Supplementary Table 3 and Movie 4). The phase shift of the maximum anisotropy of the integrin-GFP chimera relative to the angle between the membrane normal and the excitation axis in segmented leading edges, $\theta_d$, was 98.5° ± 37.6° for αL-T and 75.7° ± 46.6° for αL-F (mean ± s.d. for 16 and 35 cells, respectively). In contrast, fits of anisotropy to the $\cos^2$ function for cytosolic GFP and membrane-bound GFP (CAAX) were poor

with low amplitudes and $R^2$ values (Fig. 4e, f, Movie 5). Activating LFA-1 independently of the cytoskeleton with $Mn^{2+}$, disrupting linkage of LFA-1 to the cytoskeleton with overexpression of talin head domain, or bypassing LFA-1 engagement by using the non-integrin substrate anti-CD43 all lowered amplitude, demonstrating decreased integrin orientation with respect to the leading edge

(Fig. 4f). These results show that LFA-1 becomes oriented in the lamellipodium with respect to the adjacent leading edge, and that the extent of orientation is dependent on LFA-1 activation by a physiologic rather than exogenous mechanism, ICAM-1 engagement, and talin linkage to the actin cytoskeleton.

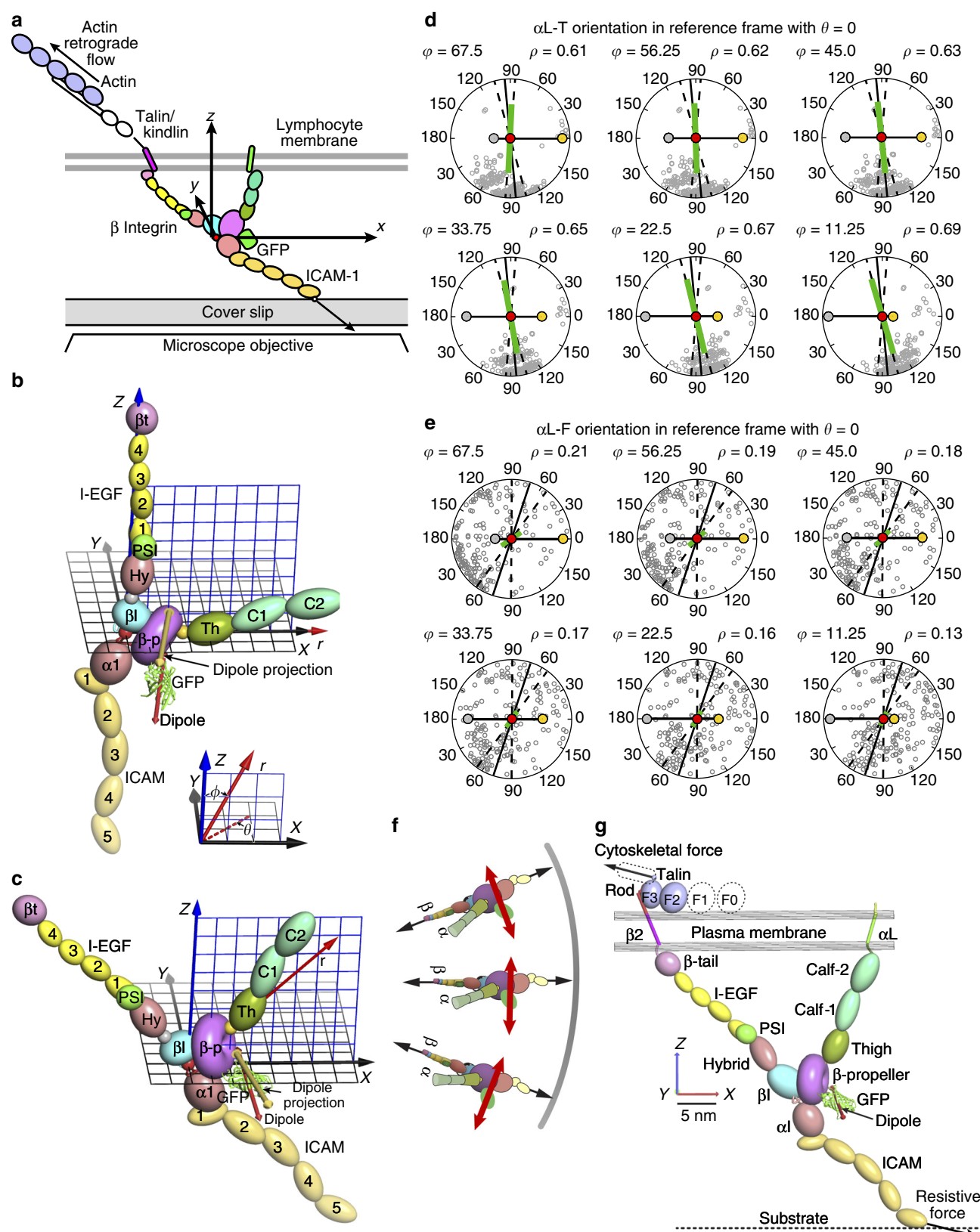

To verify and extend our measurements to higher precision, we used a second type of fluorescence microscopy. With the Instantaneous FluoPolScope[40], fluorophore dipoles are excited isotropically with circularly polarized laser TIRF excitation, and emission is split using polarization beam splitters and projected simultaneously onto four quadrants of a single CCD detector (Fig. 5a). Measurement of emission at four different angles (0, 45, 90, and 135°) enables determination of dipole orientation and polarization factor $p$ (analogous to anisotropy $r$) in each pixel with FluoPolScope. We validated the FluoPolScope with measurements on SiR-actin filaments. We found an ensemble dipole orientation of $91.4 \pm 14°$ relative to the filament axis (Supplementary Fig. 4d), in agreement our EA-TIRFM measurements.

We next measured the orientation and polarization factor of the GFP-LFA-1 transition dipole ensemble in migrating T cells. Orientation and polarization factor in regions of interest (ROI) near the leading edge or toward the cell center were calculated as the intensity-weighted sum of the values in pixels (Fig. 5b–d). Emission polarization in leading edges for αL-T was significantly higher than for αL-F and also higher than for αL-T in the cell body or for GFP in solution or in the cytoplasm (Fig. 5c). Leading edge emission polarization factor on ICAM-1 substrates was significantly decreased by extracellular $Mn^{2+}$ and was significantly lower on anti-CD43 than on ICAM-1 substrates (Fig. 5c). Integrins in ROIs near the leading edge were thus much better aligned with one another when activated physiologically than when activated by $Mn^{2+}$, and were much better aligned with one another on ICAM-1 than on CD43 substrates.

Transition dipole orientation in ROIs near the leading edge strongly correlated with orientation of the adjacent leading edge. Despite the continuous change in the membrane normal along curved leading edges of migrating cells such as those shown in Fig. 5b, e, the αL-T emission dipole remains nearly perpendicular to the membrane normal, showing it is oriented relative to actin flow (Fig. 5b, d). When absolute dipole orientation in ROIs (Fig. 5e, f) is plotted relative to the membrane normal, a narrow distribution of orientations is found (Fig. 5g). We quantitated emission dipole orientation of αL-T and αL-F relative to the membrane normal ($\theta_d$) over 264–351 ROIs from 21 to 31 cells. Orientation was $95.4° \pm 10.1°$ (mean $\pm$ s.d.) for αL-T and $71.7° \pm 17.9°$ for αL-F (Fig. 5d). These values are within 4° of those from EA-TIRFM, but have much smaller s.d., likely because phase shift calculations in EA-TIRFM were dependent on sampling a wide range of leading edge orientations and showed more cell to cell variation. Circular Gaussians also fit the FluoPolScope data well (propeller-shaped outlines in Fig. 5d). Taken together, the EA-TIRFM and Instantaneous FluoPolScope results show that in lamellipodia the transition dipole of the GFP moiety of GFP-LFA-1 fusions, and hence also the LFA moiety of these fusions, are oriented relative to the leading edge and hence relative to retrograde actin flow.

**Integrin orientation and tilt on the cell surface.** To translate GFP transition dipole orientation to the orientation of LFA-1 in the leading edge of migrating cells we utilized the orientation of the GFP excitation dipole relative to the GFP crystal structure measured by two independent techniques[28,41]. Based on the high anisotropy of GFP crystals, and the angular dependence of their polarized excitation and emission maxima, the excitation and emission dipoles of GFP are within a few degrees of one another[28,29]. The previous measurements of transition dipole orientation relative to the GFP crystal structure thus allowed us to define transition dipole orientation in GFP-LFA-1 fusion Rosetta ensembles as shown in Fig. 1e.

To determine GFP-LFA-1 fusion orientation on cell surfaces we utilized measurements of dipole orientation, and not absolute values of anisotropy or polarization factor, as the latter values can be effected by how uniformly the integrins mediating cell migration are aligned to one another and the presence of unaligned, unengaged integrins in the same regions. Absolute values of anisotropy and polarization factor will also be affected by variation in GFP orientation relative to the integrin, which is expected based on variation in orientation found in Rosetta ensembles (Fig. 1e). Furthermore in our use of transition dipole orientation, we relied more on αL-T-GFP than αL-F-GFP measurements, as Rosetta predicts a narrower range of dipole orientations and higher polarization factor for αL-T-GFP than αL-F-GFP (Fig. 1e and Supplementary Figs. 9–12, Supplementary Table 2), in agreement with its higher value of experimentally measured polarization factor (Fig. 5c). Furthermore, the experimental error in αL-T-GFP dipole orientation measurements was smaller.

Defining molecular orientation on the cell surface required constructing a frame of reference that places integrin atomic coordinates in microscope coordinates (Fig. 6a). The $xy$ plane was defined as parallel to the ICAM-coated coverslip on the inverted microscope. The $x$ direction is normal to and toward the adjacent leading edge; thus retrograde actin flow is in the $-x$ direction (Fig. 6a). The $xz$ plane was defined by three atoms with conserved positions among integrin heterodimer ligand complexes and conformational states (Fig. 6b).

The experimentally determined value for the orientation of the transition dipole for αL-T is in perfect agreement with an orientation on the cell surface of LFA-1 molecules engaged to actin and ICAM-1 of $\theta = 0°$. The experimentally determined dipole for αL-T of $\theta_d = 95.4 \pm 10.1°$, shown in Fig. 6d as a solid black line with $\pm 1$ s.d. shown with dashed black lines, aligns with the ensemble dipole predicted by Rosetta (green line) when orientation of the integrin in its reference frame is $\theta = 0°$, over a wide range of $\varphi$ tilts (Fig. 6d). This means that the integrin is oriented on the cell surface with its α-subunit–β-subunit axis parallel to the direction of actin flow as shown in Fig. 6b, c. This alignment is as predicted by the hypothesis that retrograde actin

**Fig. 6** Molecular model of LFA-1 orientation on the cell surface. **a** Schematic of the integrin-microscope reference frame. **b**, **c** Details of the integrin-microscope reference frame. $xy$ and $xz$ planes are black and blue grids, respectively. Integrin Cα atoms used to define the origin (red), $x$ axis (gold), and $xz$ plane (silver) are shown as large spheres. The GFP transition dipole (red) and its projection on the image plane (yellow–orange) are shown as cylinders with cones at each end. A spherical coordinate radial marker $r$ (red arrow) is used to compare integrin orientations between the reference state with $\theta = 0°$ and $\varphi = 90°$ **b** and integrin orientation with $\theta = 0°$ and $\varphi = 45°$ that fits data well **c**. Inset shows relation between Cartesian coordinates and spherical coordinates with $\varphi$ measured between the $z$ axis and radial marker ($r$) and $\theta$ measured between the projection of $r$ in the $xy$ plane and the $x$ axis. **d**, **e** The image plane with dipole positions of Rosetta ensemble members (lowest 40% in energy) projected from a spherical surface and shown as open gray circles for αL-T **d** and αL-F **e**. Projections are shown in the reference frame with $\theta = 0°$ and tilts in $\varphi$ ranging from 11.25 to 67.5°. Silver, red, and gold circles show the same integrin reference atoms as in **b** & **c**. The calculated ensemble transition dipole[57] is projected as a green line with length proportional to $p$. The transition dipole orientation determined by FluoPolScope is shown as black line with $\pm 1$ s.d. shown as dashed lines. **f** Schematic showing integrin and GFP dipole orientation relative to tensile force between the actin cytoskeleton and ICAM (arrows) in migrating cells. **g** Model of cytoskeletal force acting on an integrin drawn to scale tilted at $\varphi = 45°$. Structures[15,20–22,56,58–60] were assembled and rotated at domain–domain junctions known to be flexible and depicted with PyMol

flow exerts a force on the β-subunit cytoplasmic domain that is resisted by the ligand. Note that transition dipoles have a twofold symmetry axis; the transition dipole orientation we measure is therefore compatible with an integrin orientation of either $\theta = 0°$ or 180°. Nonetheless, evidence that the cytoskeleton applies the force to the integrin β-subunit forces us to choose β retrograde of α, which is fulfilled only by $\theta = 0°$. Both integrin head rotation $\theta$ in the $xy$ plane and tilt $\varphi$ relative to the $z$ axis will affect the measured projection of the GFP transition dipole in the $xy$ plane (Fig. 6b, c). The effect of tilting LFA-1 is shown by comparison of Fig. 6b ($\theta = 0°$, $\varphi = 0°$) with Fig. 6c ($\theta = 0°$, $\varphi = 45°$). Tilting changes the orientation of the transition dipole in three-dimensional space (red double-ended arrow) and its projection on the $xy$ plane, which we measure experimentally (orange doubled ended arrow) (Fig. 6b, c). Exploring a range of tilts of αL-T (variation in $\varphi$) shows that the transition dipole calculated from Rosetta ensemble members (green line in Fig. 6d) falls within one standard deviation of the experimentally determined $\theta_d$ value (black line in Fig. 6d) for $\varphi$ values between 67.5° and 22.5° (Fig. 6d and Supplementary Fig. 10). The measured $\theta_d$ value of 71.7° ± 17.9° for αL-F (solid black line, Fig. 6e) is also in perfect agreement with the calculated Rosetta ensemble transition dipole (Fig. 6e, black line); the calculated transition dipole falls within one standard deviation of the measured dipole for $\theta = 0°$ for $\varphi$ values between 45° and 11.25° (Fig. 6e and Supplementary Fig. 9). Considering that our transition dipole measurement for αL-T has an uncertainty of ± 10.1°, our results define the orientation of engaged LFA-1 on cell surfaces with respect to the reference frame as within about 10° of $\theta = 0$ and within ~25° of $\varphi = 45$. A caveat is that the average orientation that the GFP and its dipole adopt biologically with respect to the integrin may deviate from that predicted from Rosetta ensembles. However, αL-T lacks five N-terminal residues and one C-terminal residue from GFP to minimize both flexibility and uncertainty in orientation (Fig. 1d). The range of transition dipole orientations found by Rosetta is relatively narrow for αL-T (Fig. 6d) and we expect that the biological orientation is included within the range of orientations sampled by Rosetta, and not far from the predicted orientation. Our confidence in our biological result and methodology is enhanced by the independent results with αL-F. It lacks no GFP residues and has a different predicted Rosetta ensemble dipole orientation and an experimentally measured $\theta_d$ value that differs by 24° from αL-T, yet also shows that LFA-1 has an orientation in the reference frame at close to $\theta = 0$, in alignment with actin retrograde flow.

Our results suggest that on the surface of a migrating T cell, the LFA-1 head orients in the same direction as that of actin retrograde flow (Fig. 6c), and tilts relative to the membrane normal by about 45° (Fig. 6g). The integrin β-leg and ICAM-1 have flexible inter-domain junctions and the αI domain is flexibly linked to the βI domain and β-propeller domain as shown by variation among structures[2,21,22]. In contrast, the βI and β-propeller domains interact over a very large, highly stable interface and show an invariant inter-domain orientation in crystal structures. This invariance enables the GFP fusions used here to report on both βI and β-propeller domain orientation, i.e., on integrin head orientation. Force balance requires that tensile force straightens the flexible domain–domain linkages in the force-bearing chain between the actin cytoskeleton and the integrin-ligand similarly to links in a tow-chain, and aligns them in the direction of force (Fig. 6g). Thus, our measurements not only reveal the orientation of the integrin head near the center of this force-bearing chain, but also suggest that the entire chain has an orientation similar to the path that force takes through the βI domain in the integrin head (Fig. 6f).

## Discussion

We introduce a novel method for measuring molecular orientation on cell surfaces. Little has been known about cell surface receptors built from multiple tandem extracellular domains linked to single-span transmembrane domains with respect to their orientation in the plasma membrane. Furthermore, it has been difficult to relate conformational states of isolated receptor glycoproteins to their conformation, function, and orientation on cell surfaces. We have demonstrated that integrin LFA-1 becomes aligned at the leading edge of migrating T cells. Alignment is much greater at leading edges than toward the center of cells, as demonstrated by fluorescence anisotropy and polarization factor. We demonstrate that the GFP dipoles of GFP-LFA-1 fusions, and hence the LFA-1 moiety of these fusions, adopt a specific orientation with respect to the adjacent leading edge. Furthermore, with two GFP-LFA-1 fusions with distinct predicted orientations between the GFP dipole and the integrin, we define the orientation of the integrin with respect to the leading edge. This orientation is with an uncertainty of 10° exactly that predicted by the cytoskeletal force model of integrin activation, with the axis of tensile force transmission between the β-subunit cytoplasmic domain and the ligand ICAM-1 oriented in the same direction as actin retrograde flow. We obtained similar results on anisotropy/polarization factor, alignment, and orientation with two distinct fluorescent polarization microscope techniques, EA-TIRFM and Instantaneous FluoPolScope. We also made measurements of actin retrograde flow. Comparison of retrograde flow velocity on ICAM-1, CD43 antibody, and mixed ICAM-1/CD43 substrates demonstrated that LFA-1 slows actin retrograde flow in migrating T cells. The simplest interpretation of this result is that actin retrograde flow exerts a force on LFA-1. We found that flow was largely retrograde, i.e. normal to leading edge. We believe that experimental spread in our angular measurements of actin flow direction in migrating lymphocytes is due to noise, irregularity in cell shape, and the use of a method with less denoising than speckle microscopy. Measurements of actin flow with speckle microscopy in immunological synapses formed by lymphocytes have clearly demonstrated retrograde actin flow in a direction normal to the cell edge[19,31].

The simple physical concept of force balance requires that force not only orients integrins in the $xy$ plane of the membrane, but also tilts them relative to the normal to the membrane, i.e., the $z$ axis in our microscope coordinate system (Fig. 6a). Although cell surface proteins are commonly drawn in cartoons with their tandem extracellular domains normal to the membrane, as in Fig. 1a, this orientation is arbitrary; tilt has not been defined experimentally to our knowledge for the extracellular domain of any cell surface receptor. Integrin α and β-subunits each have flexible linkers between their last leg domain and transmembrane domain, and cross-linking studies and Rosetta have suggested that in the absence of applied force, integrins can adopt multiple tilts relative to the membrane[20]. Our experimental results suggest that tensile force constrains integrin tilt. The force vector applied by actin retrograde flow may be deconstructed into components parallel and normal to the membrane and it is to be expected that the parallel component is larger. Indeed, super-resolution measurements on talin show that it tilts in focal adhesions at an angle of ~75° with respect to the membrane normal[42]. We measured here a tilt ($\varphi$ angle) of ~45 ± 25° for LFA-1. Orientation in the $xy$ plane ($\theta$) is much better determined than tilt for αL-T because the orientation of the dipole in three dimensions is close to the $xy$ plane; thus, $\varphi$ has much less effect than $\theta$ on the experimental measurable of the projection on the $xy$ plane of the dipole (compare red dipole and its gold projection at $\varphi = 0$ and 45° in Fig. 6b, c, respectively).

Our results form an important bridge between studies of forces associated with integrin adhesion and migration and structural studies on integrins. Single molecule forces measured on integrin ligands and several actin cytoskeletal adaptors have emphasized the importance of integrins in transmitting force between the extracellular environment and the cytoskeleton[7,8]. Recently, forces have also been measured within integrin cytoplasmic domains in T cells migrating on ICAM-1 substrates[9]. Force is exerted on the cytoplasmic domain of the LFA-1 β-subunit, but not the α-subunit, and is dependent on binding to ICAM-1 on the substrate and intact binding sites in the β-subunit cytoplasmic domain for talin and kindlin, which couple integrins to the actin cytoskeleton. We have measured these forces as largely between 1 and 2 pN in the LFA-1 β-subunit cytoplasmic domain[9]; similarly, forces on integrin ligands have been measured as largely within the 1–3 pN range[8]. Our results here show that the forces exerted on integrins are sufficient to align them, and that the specific orientation found for integrins at the leading edge of migrating cells is consistent with the orientation predicted for force application by the cytoskeleton to the integrin β-subunit cytoplasmic domain that is transmitted through the integrin and resisted by a ligand bound to a substrate.

In contrast to force measurements in cells, crystal, EM, NMR, SAXS, and neutron scattering structures of integrins are determined in the absence of force. Elegant structures have revealed intact integrins, integrin ectodomains and their complexes with ligands, integrin transmembrane domains, and integrin cytoplasmic domains and their complexes with intracellular effectors that link or inhibit linkage to the cytoskeleton[2,6,11,12,43,44]. The demonstration here that we can use fluorescence microscopy to define a specific orientation for integrin atomic structures on the surface of migrating cells now places these structures not only in the context of intact functioning cells, but makes it essential to discuss integrin structural biology in the context of force application by actin retrograde flow. For example, flexibility between many of the domains in integrins enables them to straighten and align with their domain–domain junctions parallel to the force. Flexibility in poorly structured residues that link the ectodomain to the plasma membrane enables integrin tilting[20] as suggested by our measurements here. Moreover, the integrin β-subunit transmembrane domain tilts when separated from the α-subunit transmembrane domain, as occurs upon integrin activation[2,11,12], consistent with the tilt suggested here.

Although almost every paper on an integrin complex discusses how binding of a ligand or effector may regulate integrin activation by selecting among integrin conformational states, the implications of complex formation for tensile force transmission from the cytoskeleton that could select among integrin conformational states is discussed by few workers in the field. Our findings on integrin orientation on cell surfaces provide strong support for the cytoskeletal force model of integrin activation[2,3,9,13–15]. Recent thermodynamic measurements and calculations provide further insights by showing that force enables ultrasensitive regulation of integrin adhesiveness[5,16]. Among the three integrin conformational states shown in Fig. 1a, only the extended-open state has high affinity for ligand[2,5]. This affinity has been measured as ~1000-fold higher for integrin LFA-1[3] and more precisely as 4000-fold higher for integrin α5β1 for the extended-open than the bent-closed and extended-closed conformations[5]. Measurements of free energies of integrin conformational states on the cell surface have shown that the bent-closed conformation is substantially lower in free energy than the extended-closed or extended-open conformation, with >99% of integrins in the bent-closed conformation at equilibrium[5]. Binding of adaptors such as talin has little effect on this equilibrium[16]. However, tensile force stabilizes equilibria by a potential energy equal to the force times the difference between each state in extension along the force-bearing axis. Thermodynamic calculations show that in an equilibrium model in which integrins are allowed to bind intracellular adaptors and extracellular ligands, and tensile force is applied when both adaptor and ligand bind, the extended-open conformation is greatly stabilized, and becomes the predominant state. Force in the same range as measured on integrins and ligands, in the 2 pN range, is sufficient to fully activate integrins. Moreover, whereas adaptors such as talin have little effect in the absence of force, they provide ultrasensitive integrin activation in the presence of force[16].

Our fluorescence measurements here are for the ensemble of all three conformational states of our GFP-LFA-1 fusions. Based on thermodynamic measurements and calculations, essentially all integrins that are simultaneously bound to a force-generating cytoskeleton and to ICAM-1 are in the extended-open conformation, whereas integrins that are unbound or bound only to adaptors or ICAM-1 are predominantly in the bent-closed conformation[16]. Only the adaptor and ligand-engaged, extended-open integrins can be aligned by actin retrograde flow and contribute to the alignment measured as anisotropy and polarization factor and the orientation measured as dipole orientation here. Unengaged, bent-closed integrins will be randomly oriented on the cell surface, are likely to make up a substantial and perhaps the majority of the integrin ensemble on the cell surface both toward the center of cells and at the leading edge, and contribute to lowering observed anisotropy and polarization factor values.

Stabilizing high affinity of integrins for extracellular ligand by force application by the actin cytoskeleton provides a simple and elegant mechanism for coordinating cytoskeletal activity inside the cell with binding to ligands in the extracellular environment during cell migration[9,15]. The mechanisms that regulate actin cytoskeleton dynamics are highly complex, and the coordination by tensile force transmission of ligand binding with actin cytoskeletal motion not only avoids the need to have a regulatory mechanism for integrin activation that duplicates actin regulatory mechanisms but also enables efficient coordination between the actin cytoskeleton and integrins to provide cellular traction precisely where cytoskeletal force is exerted. Thus, our results place molecular understanding of integrin structure and function within the context of integrin function in cell migration and in linking the extracellular environment to the actin cytoskeleton. As 22 of the 24 mammalian integrin αβ heterodimers have β-subunits that link to the actin cytoskeleton, the findings here with integrin LFA-1 are expected to be of wide relevance among integrins. Indeed, a paper submitted concurrently with ours to BioRxiv demonstrates that integrin αVβ3, which is distant from LFA-1 in the integrin family and differs in the respects that it recognizes the ligand fibronectin, lacks an αI domain, is found in focal adhesions, and is linked to slow actin stress fiber retrograde flow and slow migration, also is aligned by actin flow[45,46].

## Methods

**Integrin-GFP constructs**. EGFP or moxGFP[47] were inserted into the β3-β4 loop of blade 4 of the αL integrin β-propeller domain[26]. Integrin Gly residues adjacent to GFP were mutated to Ala or Gln residues for helix propensity as indicated, linkers were added for flexibility, or GFP residues were deleted for less flexibility as follows. N- and C-terminal insertion sites are shown with residues for integrin WT or mutant sequences in plain text, linkers in bold, and GFP italicized: αL-GFP-F, EPQG *MVSKGEELF…MDELYK* GGHW; αL-GFP-L, EPQG**SGSG** *MVSKGEELF…MDELYK* **GSGS** GGHW; αL1, EPQA *EELF…MD* AQHW; αL2, EPQA *EELF…MDE* AQHW; αL3, EPQA *EELF…MDEL* AQHW; αL4, EPQA *ELF…MDELY* AQHW; αL5, EPQA *LF…MDELY* AQHW. The αL3 construct worked best during functional testing of αL1- αL5 and was used throughout this study with the name αL-GFP-T.

Integrin α and β-subunit cDNA were made using three-segment (A,B,C) overlap PCR with wild-type human αL cDNA and either pEGFP-N1 (Clontech) or moxGFP[47] (for αL-GFP-T) as sources for GFP cDNA. After the three segments had

been made and stitched together through PCR (Accuprime Pfx, high-fidelity polymerase, ThermoFisher), the complete A–C sequence and the wild-type αL-pcDNA3.1 plasmid were cut with restriction enzymes (New England Biolabs) and ligated together with T4 ligase (Roche) after dephosphorylation (rAPID alkaline phosphatase, Roche) and purification (Qiagen) of the linearized plasmid. The overall plasmid integrities were verified by size matching of multi-site single restriction enzyme digestion compared to virtual digest patterns (Serial Cloner) and the inserts were verified by full sequencing. Surface expression of the αL-GFP constructs was validated by transient co-expression with β2 in 293 T cells. For αL-GFP-F the primers used were: A1: 5′-AGA TGT GGT TCT AGA GCC ACC ATG AAG GAT TCC TGC-3′; A2: 5′-TGA ACA GCT CCT CGC CCT TGC TCA CCA TGC CCT GTG GCT CTT GGA AC-3′; B1: 5′-AGT GCT GCT GTT CCA AGA GCC ACA GGG CAT GGT GAG CAA GGG CGA G-3′; B2: 5′-ATG GAT TGT CTG GAC CTG GCT CCA GTG TCC TCC CTT GTA CAG CTC GTC CAT GCC-3′; C1: 5′-ATC ACT CTC GGC ATG GAC GAG CTG TAC AAG GGA GGA CAC TGG AGC CAG-3′; C2: 5′-ACT CTT AGT AGC GGC CGC TCA GTC CTT GCC ACC ACC-3′. Primers for αL-GFP-L were: A1: 5′-AGA TGT GGT TCT AGA GCC ACC ATG AAG GAT TCC TGC-3′; A2: 5′-AGC TCC TCG CCC TTG CTC ACC ATG CCA GAT CCA GAG CCC TGT GGC TCT TGG AAC-3′; B1: 5′-AGT GCT GCT GTT CCA AGA GCC ACA GGG CTC TGG ATC TGG CAT GGT GAG CAA GGG CGA G-3′; B2: 5′-TGT CTG GAC CTG GCT CCA GTG TCC TCC GCT GCC TGA GCC CTT GTA CAG CTC GTC CAT GCC-3′; C1: 5′-ATC ACT CTC GGC ATG GAC GAG CTG TAC AAG GGC TCA GGC AGC GGA GGA CAC TGG AGC CAG-3′; C2: 5′-ACT CTT AGT AGC GGC CGC TCA GTC CTT GCC ACC ACC-3′. Primers for αL-GFP-T were: A1: 5′-AGA TGT GGT TCT AGA GCC ACC ATG AAG GAT TCC TGC-3′; A2: 5′-CAC CAG AAT AGG GAC CAC TCC AGT AAA CAG TTC CTC AGC CTG TGG CTC TTG GAA CAG CAG-3′; B1: 5′-GGC CGA GTG CTG CTG TTC CAA GAG CCA CAG GCT GAG GAA CTG TTT ACT GGA GTG GTC CC-3′; B2: 5′-CCA TGG ATT GTC TGG ACC TGG CTC CAG TGT TGA GCA TAC AGC TCA TCC ATT CCG TGG GTG-3′; C1: 5′-GCT GCT GGA ATC ACC GAC GGA ATG GAT GAG CTG TAT GCT CAA CAC TGG AGC CAG GTC CAG-3′; C2: 5′-ACT CTT AGT AGC GGC CGC TCA GTC CTT GCC ACC ACC-3′. See Supplementary Table 1 for details on amino acid sequence and constructs used for simulations. Other constructs used were: Lifeact[48] fused with mCherry or mNeonGreen, talin head fused to mApple, and mApple with CAAX sequence added.

**Reagents.** Wild-type soluble ICAM-1-His6 (D1-D5) was expressed in 293 cells and purified on Ni-NTA agarose[3] and purified on Ni-NTA agarose. Human SDF1-α was from R&D System. Cytochalasin D was from Santa Cruz. Blebbistatin was from AbCam. Phalloidin-Alexa488 was from Invitrogen. Anti-CD43 was from ebiosciences. Glass-bottom dishes and plates were from Mattek. Leibovitz's L-15 medium and RPMI-1640 medium were from Life Technologies. The reagents for the lentiviral Gateway system were from Life Technologies. Nucleofector Kit V was from Lonza.

**Cells.** Jurkat T cells (ATCC clone E6.1) were cultured in RPM1-1640 medium with 10% FBS in 5% $CO_2$ and supplemented with 3 µg ml$^{-1}$ puromycin and/or 1 µg ml$^{-1}$ blasticidin if they had been lentivirally transduced.

**Lentiviral transduction of cells.** The Gateway system from Invitrogen was used to create lentiviral constructs. The integrin constructs were inserted either into pLX302 or pLX304. Virus was produced in 293 T cells by co-transfecting the lentiviral plasmids with psPAX2 and CMV-VSV-G. Virus in supernatants was concentrated using Lenti-X (Clontech). Jurkat cells were transduced and selected using puromycin (pLX302) or blasticidin (pLX304).

**Live imaging.** Glass-bottom dishes or plates were adsorbed overnight at 4 °C with 10–20 µg ml$^{-1}$ ICAM-1 in carbonate buffer (pH 9.6), followed by blocking at 37 °C with 1% BSA in L-15 medium for 30–60 min, and washing with base imaging media consisting of L-15 supplemented with 2 mg ml$^{-1}$ glucose. Cells were suspended in base medium supplemented with 100 ng ml$^{-1}$ SDF1-α. Before imaging, cells were added to the dish or well on the microscope held at 37 °C and allowed to settle.

**Fixed cell imaging.** Cells were prepared as for live imaging and allowed to migrate at 37 °C for 30 min. Inhibitors were added for inhibitor-specific times prior to fixation: DMSO, 1:2000, 30 min; cytochalasin D, 100 nM, 15 min; blebbistatin, 100 µM, 30 min. Fixation with an equal volume of paraformaldehyde at a final concentration of 2% was for 10 min at 37 °C. After washing with phosphate-buffered saline, cells were imaged as for live samples.

**Actin purification, polymerization, labeling and glass fixation.** Actin was purified from chicken breast following the protocol from Spudich et al.[49]. The monomeric form was maintained in G-buffer (2 mM Tris Base, 0.2 mM ATP, 0.5 mM TCEP-HCl, 0.04% NaN$_3$, 0.1 mM CaCl$_2$, pH 7.0) on ice. For actin polymerization, the G-actin was mixed with G-buffer and 10% v/v of 10× ME buffer (100 mM MgCl$_2$, 20 mM EGTA, pH 7.2) to obtain an actin concentration of 10 µM and incubated for 2 min to replace G-actin bound Ca$^{2+}$ ions with Mg$^{2+}$. Next an equal amount of polymerization buffer was added to induce F-actin polymerization at a final actin concentration of 5 µM in KMEH (50 mM KCl, 2 mM MgCl$_2$, 1 mM EGTA, 20 mM HEPES, pH 7.2) supplemented with 2 mM ATP and 1 mg/ml BSA. After 20–30 min incubation, the F-actin was labeled by addition of 500 nM Phalloidin-Alexa488 (Invitrogen) and/or 1 µM SiR (Cytoskeleton Inc) and incubation of 10 min at room temperature. Then, F-actin was sheared by pipetting up and down 10 times, diluted in KMEH and transferred to a 0.01% Poly-L-Lysine coated glass-bottom dish at final concentration of 10 nM. After 15 min of incubation, unbound actin filaments were washed away with KMEH buffer and the samples were imaged in a 100 × 1.49 NA TIRF microscope. For Poly-L-Lysine coating, 0.01% PLL was aseptically coated onto the surface of no. 1 glass coverslips and rocked gently to ensure even coating. After 5 min, the excess solution was removed by aspiration and the surface was rinsed with tissue culture grade water and left drying under laminar flow for at least 2 h before use.

**EA-TIRFM.** EA-TIRFM images were acquired using the TIRF mode on a Nikon Eclipse TiE inverted microscope equipped with a motorized TIRF illuminator (Nikon, USA) and a motorized stage (TI-S-ER motorized stage with encoders; Nikon, USA) fed by a multi-wavelength (405 nm (15–25 mW), 488 nm (45–55 mW), 561 nm (45–55 mW), 640 nm (35–45 mW)) polarization-maintaining fiber coupled monolithic laser combiner (Model MLC400, Agilent Technologies). This arrangement generates a polarized TIRF evanescent field at the sample plane[27].

Images were collected with a fixed magnification using a ×100 Plan Apo 1.49 NA TIRF objective (Nikon, USA) fitted with a Perfect Focus System (PFS3; Nikon, USA) and a ×1.5 tube lens to yield a final pixel size corresponding to 109 nm. The typical TIRF illumination depth using 488 nm was 150–200 nm. Band-pass emission filters (ET525/50, ET600/50, and ET700/75; Chroma Technology Corp, USA) were mounted onto a motorized turret below the dichroic mirror (405/488/561/638 TIRF Quad cube; Chroma Technology Corp, USA).

Emission from the polarized evanescent TIRF field was split into constituent $p$ and $s$-polarized components using a high performance nano-wire grid polarizing beam splitter (TR-EMFS-F03; Moxtek Inc., USA).The resulting parallel and perpendicular components were imaged with separate, orthogonally placed iXon Ultra 897 EMCCD cameras (Andor Technology, Belfast, Northern Ireland) using the TuCam two-camera imaging adapter (C-Mount Version [S-CMT]; 1× Magnification [TR-DCIS-100]; Andor Technology, Belfast, Northern Ireland). Images were acquired using the Nikon Imaging Software (NIS Elements Advanced Research; Nikon, USA) with a dual-camera plugin using the electron multiplying (EM) gain mode.

**Instantaneous FluoPolScope.** A custom microscope using opto-mechanics from Newport Corp was built on an optical table. Laser beams (Coherent Sapphire 488 nm, 20 mW and Melles-Griot 561 nm, 25 mW) were routed through custom optics and focused on the back focal plane of a 100 × 1.49 NA objective (Nikon ×100 ApoTIRF 1.49 NA). The objective was placed on a Piezo Z-collar (PIP-721 PIFOC) for precise focusing. Laser beams were circularly polarized using a combination of a half wave plate and quarter wave plate (Meadowlark Optics). To achieve isotropic excitation within the focal plane and along the optical axis of the microscope, the circularly polarized laser beam was rapidly rotated (300–400 Hz) in the back focal plane of the objective with a large enough radius to achieve total internal reflection at the specimen plane. Dual-band dichroic mirror (Semrock Di01-R488/561) was used to separate laser lines (reflected) and emissions corresponding to GFP and mCherry (transmitted). The specific emission channel was selected using bandpass filters mounted in a filter wheel (Finger Lakes instruments). A quadrant imaging system as described in Mehta et al.[40] was used for instantaneous analysis of fluorescence emission along four polarization orientations at 45° increments (I0, I45, I90, I135). Dual-channel imaging of live cells was performed using Micro-Manager (version 1.4.15). All images were acquired using an EMCCD camera (Cascade II: 1024; Photometrics, Tuscon, AZ) operated in the 5 MHz readout mode with EM gain.

**System effects on anisotropy.** Fluorescence anisotropy is dependent on the experimental set up as well as properties of the fluorophore such as: (1) angular diffusion coefficient of tethered fluorophores (2) degree of alignment to the polarization axis when fluorophores experience rotational constraint, and (3) dominant orientation of fluorophores when they are aligned. The high numerical aperture used in TIRF-based imaging introduces polarization mixing, reducing the range of anisotropy magnitude from 1 to −0.5 to 0.8 to −0.4 (Supplementary Fig. 1). For GFP in solution, these effects result in a drop of the observed anisotropy from the typical value of 0.32[50] to ~0.18–0.20[27,51]. As these depolarization effects are similar across the various experimental systems used in this article, the relative changes in anisotropy and not the absolute values of anisotropy are significant.

**SIM.** 3D-SIM data were collected on a DeltaVision OMX V4 Blaze system (GE Healthcare) equipped with a 60x/1.42 N.A. plan Apo oil immersion objective lens (Olympus), a 488 nm diode laser, and an Edge 5.5 sCMOS camera (PCO). Image stacks of ~2–3 µm (fixed) or ≤1 µm (live) were acquired with a z-step of 125 nm and with 15 raw images per plane (five phases, three angles). Spherical aberration was minimized using immersion oil matching[52].

## Image processing and analysis

*EA-TIRFM.* Image processing and analysis was mainly carried out using MATLAB 2014a. Functions handling all steps of the image processing were developed into a semi-automatic software package with optional manual steps for image registration. Stepwise, images were imported from the original files and sorted into channels; all metadata were extracted and saved; image registration was carried out with one of three options: manual reference image of submicron beads initialization, automatic reference image initialization or automatic registration using cell images; G factors were calculated daily based on fluorescein solution images; images were G factor corrected; and background was masked by thresholding at a value 3 s.d.s above background, where the background intensity distribution is estimated by fitting the "left half" of a Gaussian function (the portion below its mean) to the left shoulder of the image intensity histogram. This mask was then used to find and subtract the average background intensity on a frame-by-frame basis for each channel. For all anisotropy calculations the data was pre-filtered with a $3 \times 3$ intensity-weighted average applied to all pixels. To minimize artifacts from division of small integers, only pixels with intensities above 4 times the background standard deviation of the current frame were used; finally, anisotropy was visualized using a heat map. Intensity-weighted anisotropy ($r$) of each group of nine pixels was calculated and displayed in the central pixel through the relationship:

$$r = \frac{I_{\parallel} - I_{\perp}}{I_{\parallel} + 2 \cdot I_{\perp}} \quad (1)$$

In this equation, the difference between parallel and perpendicular intensity is divided by total intensity where perpendicular intensity is added twice to account for the two planes perpendicular to the parallel plane in three-dimensional fluorescence emission.

For image analysis the initial step was to identify cells. In brief, as multiple cells could be present in the same image and have different fluorescent expression, the background masks described above was used to initialize cell segmentation independently for each potential cell area. These regions were slightly expanded (five-pixel dilation operation) to make sure that background was included to maintain contrast for increased robustness of the algorithm. Either active contour segmentation (energy minimizing that separates between foreground and background) or intensity distribution-based threshold segmentation (similar to the background masking described above) was used to produce an initial cell mask. Mathematical morphology (a closure operation with a radius of one pixel, small object removal, and filling of holes) was applied to further refine these masks, producing accurate cell outlines. For live imaging data, a four-dimensional bounding box (3D space plus time dimension of at least five frames) was used to make sure that cells were consistently segmented during the movie. For edge segmentation, the cell mask was eroded by 10 pixels, and then inversely combined with the original mask to generate an edge mask. For protrusion detection, the difference between the cell masks in neighboring frames were evaluated with a four-dimensional bounding box (positive area of 2000 pixels and at least five frames) and were stored as individual protrusion masks. Leading edge segmentation was carried out by combining edge and protrusion masks for positive protrusions. Signal-to-noise ratio (SNR) was determined for each segmented region,

$$SNR = \frac{\mu}{\sigma} \quad (2)$$

where $\mu$ is the mean intensity of the perpendicular channel and $\sigma$ is the standard deviation of the background intensity in that channel. Any region with a SNR lower than five was excluded from analysis (Supplementary Fig. 2c). Given the variable cell shapes, especially between frames and in protrusive regions, an orientation mapping algorithm was devised that would assign relative orientation values in a reproducible manner. It is based on calculating the vector away from the edge for each pixel in cell masks. For circular cells, it yields an orientation axis of 0–180°, that falls along the polarization axis with clockwise assignment of orientation values (see simulation in Supplementary Fig. 8d). For irregular shapes such as polarized cells, the orientation assignment is not always comparable across cells, only within. This also means that for non-circular objects the orientation values are not absolutely correlated with the polarization axis and introduces variation in phase shift estimates across cells, but with consistent angular dependency estimates (amplitude). The cell masks were smoothed with a three-pixel radius closure operation followed by an Euclidian distance transform of the inverse mask,

$$distance = \sqrt{(x_1 - x_2)^2 + (y_1 - y_2)^2} \quad (3)$$

with $x$ and $y$ pixel coordinates. A numerical two-dimensional gradient is calculated from the distance transform,

$$gradient\,(F(x,y)) = \frac{\delta F}{\delta x}\mathbf{i}^{\wedge} + \frac{\delta F}{\delta y}\mathbf{j}^{\wedge} \quad (4)$$

and the inverse tangent is used to return the relative orientation value for each pixel. To assess angular dependence, the orientation values were binned to the

closest 10 or 15 degrees and then the orientation and anisotropy values for each pixel were fitted to a $cos^2$ function,

$$r = C + A cos^2(\gamma - \theta_D) \quad (5)$$

and the absolute amplitude of this is used as a measure to report the degree of angular dependence. The angle between the membrane normal and the excitation axis ($\gamma$), is defined in the counter-clockwise direction, as is the angle between the transition dipole and the membrane normal ($\theta_d$), equivalent to a phase shift in the cosine function. To verify whether it is possible to obtain a correct $\theta_d$ from assigning orientations to an anisotropy map, and fitting these to Equation 5, simulated images with known variables were used (Fig. 4b). In all cases, the fits were perfect ($R^2 = 1.00$), and the correct amplitude and phase shifts were found.

To get an alternative measure of angular dependence, a Fourier-based analysis (Fast Fourier Transform) was used on the running average of orientation relative to anisotropy (Supplementary Figs. 5–8). The DC term was removed by subtracting the mean anisotropy value and the amplitude value from the second bin (or first frequency, equivalent to a basic wave) was recorded.

*Instantaneous FluoPolScope.* Image analysis was performed using custom code developed in MATLAB 2014a. Algorithms are available upon request. The four polarization-resolved quadrants of the integrin-GFP channel were cropped, registered, and throughput-normalized as described in ref. [40]. We calibrated the differences in throughput of four polarization-resolved quadrants using FITC solution and spatial transformations required to register them using fluorescent beads[40]. The total intensity image of each cell was segmented along the leading edge into $1000 \times 500$ nm masks with the long axis tangential to the membrane. The cell body was segmented using masks with the same long axis angle relative to the microscope frame of reference, to ensure no effect of segmentation orientation, and distributed around the cell body >1000 nm from the leading edge. Segments were digitized, and membrane normal (yellow lines in Fig. 5b) computed as perpendicular to the segment long axis. Background polarization and excitation imbalance were determined for each cell from an approximately 0.28 μm square ($4 \times 4$ pixels) within the center of each cell. Background-corrected polarization-resolved intensities (I0, I45, I90, I135) were then summed over each segment. These sum intensities per segment were used to compute dipole orientation ($\theta$) and polarization factor ($p$) per segment as follows:

$$\theta_D = \tan^{-1}\frac{I0 - I90}{I45 - I135} \quad (6)$$

$$p = \frac{\sqrt{(I0 - I90)^2 + (I45 - I135)^2}}{0.5 \times (I0 + I45 + I90 + I135)} \quad (7)$$

For each segment, GFP dipole orientation relative to the membrane normal, $\theta_d$, was calculated as the angle in the counter-clockwise direction from the membrane normal.

*SIM.* Super-resolution images were computationally reconstructed from the raw data sets with a channel-specific measured optical transfer function (OTF) and a Wiener filter constant of 0.002–0.003 using softWoRx 6.1.3 (GE Healthcare). Live-cell data sets were collected at 37 °C using objective and stage-top heaters (GE Healthcare) and fixed data sets were collected at room temperature. Live-cell image acquisition was optimized to keep peak laser intensity below ~30 W cm$^{-2}$ and exposure times below 10 ms (≤2 s per Z-stack) to minimize motion-induced reconstruction artifacts.

Actin flow velocity and direction was determined using the Flow-J optical flow analysis[32] plugin within the Fiji image processing package[53,54]. The Lucas and Kanade algorithm[55] was applied to each live structured illumination time series to calculate the velocity vector for each pixel in each frame. The cell was segmented along the leading edge into $1000 \times 500$ nm masks with the long axis tangential to the membrane and velocity vectors averaged within each mask to determine actin flow speed and angle relative to the microscope frame of reference $X$ axis. Concurrently, the angle of the membrane tangent of each segment was measured. The angles were subtracted to determine the angle of actin flow relative to the cell membrane. Mean actin velocity was calculated and conditions compared for statistical difference using the Mann–Whitney test.

Kymographic analysis to determine actin flow velocity was performed as described in Comrie et al.[34], with SIM movies of T cells expressing Lifeact-mNeonGreen being analyzed using the Fiji image processing package to generate a vertical kymograph traversing the cell leading edge and lamellipodium. The flow rate was calculated based on the slope of deflection of F-actin from the vertical direction[34] (Supplementary Fig. 3b).

**Estimation of GFP dipole orientation relative to integrin.** Low-energy orientations between the inserted GFP and integrin were efficiently sampled using Rosetta. Rosetta found low-energy conformations for "loop" sequences at the two integrin-GFP junctions. The remainder of integrin was rigid and GFP was rigid except for

solvent-exposed sidechains. Conformations of the junction loops were found that permitted loop closure and prevented rigid body clashes followed by further loop relaxation and sidechain optimization to minimize energy[23]. Integrin sequences within two residues of the insertion site, linker residues, and GFP residues that vary in position or are disordered in GFP structures (residues 1–6 (MVSKGE) and 229–238 (ITLGMDELYK) were included in the loop regions that were subjected to backbone and sidechain optimization. Further adjacent residues were subjected to sidechain optimization only (highlighted in gold, Fig. 1d). For computational efficiency, only β-propeller residues 330–483 were included in the $\alpha_L\beta_2$ model; other domains of the α-subunit and the β-subunit were too distal to clash with GFP, as confirmed when full length models were subsequently built to model the GFP–$\alpha_L\beta_2$ fusions. The αI domain can vary markedly in orientation relative to the β-propeller domain in which it is inserted; therefore, we chose the most physiologic orientation available, from an $\alpha_X\beta_2$ ectodomain crystal structure in which the internal ligand of the αI domain binds to a pocket at the β-propeller interface with the βI domain, concomitantly with activation of the high affinity, open conformation of the αI domain[21]. A complete LFA-1 ectodomain model was built by superimposing the β-propeller domain from an $\alpha_L\beta_2$ headpiece crystal structure[22] and the αI domain from a complex of its high-affinity state with ICAM-1[56] onto the cocked $\alpha_X\beta_2$ crystal structure[21].

For each integrin-GFP fusion construct, Rosetta output an ensemble of structures that effectively sampled low-energy GFP-integrin orientations. Longer loop junctions typically enabled a larger number of loop closures enabling larger ensembles; however, no closures were found for αL5, suggesting its loops were too short. The ensemble of GFP orientations was visualized by superimposition on the integrin in the integrin-microscope reference frame. Dipole orientations were calculated in spherical coordinates. Models were ranked according to total energy and compared for angular distribution. The two lower energy quintiles had a more restricted range of dipole orientations; therefore we used the 40% lowest energy models as the ensemble for calculations of ensemble dipole orientation and polarization factor.

The first chemically plausible orientation (within the planar ring system of the amino acid residues that fuse to form the fluorophore) for the GFP excitation dipole was determined from the polarized light absorption spectra by GFP crystals and had to account for the four distinct GFP molecules present with different dipole orientations in the crystal lattice and four mathematically possible solutions to the equations used to define excitation dipole orientation[28]. Subsequently, the equations were corrected and the solution refined[41]. In addition, it was realized that previously reported visible pump/IR probe measurements could be used to calculate the orientation of the fluorophore transition dipole relative to carbonyl stretch vibrational transition moments that are well defined spectroscopically and assigned to carbon-oxygen bonds between atoms well characterized for their positions in GFP crystal structures[41]. The transition dipole orientations calculated by these two independent methods were in good agreement, and authors of the latter publication (X. Shi and S.G. Boxer) kindly provided the transition dipole orientation for GFP chain B in the coordinate system of the high-resolution structure PDB ID code 1w7s as a line with slope $x = -0.026$, $y = 0.871$, and $z = 0.439$. A line with this slope drawn through the hydroxyl oxygen atom of the chromophore closely matched a line with $\alpha = 6.5°$ in Fig. 6 of Shi et al.[41] and was approximated in integrin-GFP ensembles as a line drawn through the Val-112 $N$ atom and the average of the positions of the Asn-146 C and Ser-147 O atoms in the GFP moiety. Rosetta ensemble transition dipole orientations and polarization factors were calculated from the differences and ratio of simulated intensities as described in the methods section.

**Integrin orientation on the cell surface**. In the extended-open conformation in which integrins bind ligands, the interface of the α-subunit β-propeller and β-subunit βI domains, which form the head and bind external ligands (as with $\alpha_V\beta_3$) or internal ligands (as with $\alpha_L\beta_2$), faces away from the integrin legs, which connect to the transmembrane and cytoplasmic domains. To orient the liganded integrin in this manner, we first skeletonized the ligand-bound integrin into three key Cα atom points. The ligand-point is at the strongly bound Asp of RGD or internal ligand Glu of αI integrins. Two junction points are near sites of pivoting movements between the head and legs, yet are sufficiently inward in the head to show little variation in position among independent integrin-ligand complexes or among conformational states, and are sufficiently conserved in position to enable comparison among distinct integrin heterodimers. Furthermore, these points move little in SMD stimulations[15]. The α-junction point is at the C-terminus of the α-subunit β-propeller domain (Arg-438 in $\alpha_V$, Gln-451 in $\alpha_{IIb}$, Arg-588 in $\alpha_L$, or Arg-597 in $\alpha_X$) that connects to the flexible thigh domain. The β-junction point is at the N-terminus of the βI domain (Pro-111 in $\beta_3$ or Pro-104 in $\beta_2$) that connects to the hybrid domain. These junction points are between β-strands in adjacent domains, each of which are central strands in their β-sheets and are thus highly force-resistant.

Cartesian and spherical coordinate reference frames were defined to enable dipole orientations measured in microscopes, the orientation of the transition dipole in GFP, and the orientation of GFP with respect to the integrin in Rosetta ensembles to be used to define integrin orientation on cell surfaces. The coordinate system is defined by the three Cα atoms in the integrin and ligand described in the

preceding paragraph. Because force passes through the junction between the ligand and integrin, and force balance requires that tensile force cause them to pivot toward alignment with the direction of force exertion, the ligand-point is used to define the origin. The line between the ligand point and the α-junction point defines the X axis. The β-junction point defines the XZ plane and lies near the Z axis. These positions in turn define the XY plane, which lies parallel to the microscope image plane and the plasma membrane of the cell adhering through integrins to ligands on the substrate.

As the reference frame is constructed with the ligand and head junction points in the XZ plane and optimally orients the integrin head toward the substrate, the integrin will remain close to this plane when it is tilted by cytoskeletal force, as shown by SMD[15]. Spherical coordinates are useful for defining the orientation of the integrin and GFP transition dipole relative to the Cartesian reference frame. X, Y, Z positions in Cartesian coordinates are defined in spherical coordinates with the radial distance r and angles $\theta$ and $\varphi$. Integrin and dipole orientations are defined relative to the X axis with the α-junction point and r lying on this axis such that $\theta = 0°$ and $\varphi = 90°$. The orientation of r is defined by its angle $\varphi$ with the Z axis and its angle $\theta$ with the X axis when projected on the XY plane. Similarly, the orientation of the GFP transition dipole when projected on the microscope imaging plane in our fluorescence microscopy experiments is measured as $\theta_d$ relative to the direction of lamellipodial movement in the direction $\theta = 0°$ (normal to the leading edge). The reference frame is such that when actin retrograde flow is in the direction $\theta = 180°$, the traction force model for integrin activation predicts that (1) the line between the ligand and the α-junction points tilts toward the Z axis with a decrease in $\varphi$ and (2) the projection of this line on the XY plane has $\theta = 0°$, as in the reference frame.

**Data availability**. The data that support the findings of this study are available from the first author or corresponding author upon reasonable request.

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

## Acknowledgements

We thank Nikon Instruments and Andor Technology for use of imaging equipment, Thomas Holder at Schrodinger for Pymol/Python scripts and advice to create cgo representations of ellipsoids, torus, and dipoles, and Einat Schnur, Gabriel Billings, Darius Vasco Köster, and Amy Gladfelter for help, advice, and discussion. Supported by the Lillie Research award from Marine Biological Laboratory and the University of Chicago (C.M.W., T.A.S., S.M., T.T.), NIH 5R13GM085967 grant to the Physiology Course at Marine Biological Laboratory, HHMI Summer Institute at Marine Biological Laboratory (S.M.), NIH CA31798 (T.A.S., P.N., T.I.M.), NIH GM100160 (T.T., S.M.), NIH GM092802 (D.B., N.K.), NIH GM114274 (R.O., S.M.) National Center for Biological Sciences-Tata Institute of Fundamental Research (S.M., J.M.K.), J.C. Bose Fellowship and HFSP Grant RGP0027/2012 (S.M.), NHLBI Division of Intramural Research (C.M.W., V.S.), Swedish Research Council VR 524-2011-891 Fellowship (P.N.), Swedish Society for Medical Research SSMF Fellowship (P.N.), Crafoord Foundation (P.N.).

## Author contributions

This project was initiated at the Woods Hole Physiology Course. P.N., T.I.M., S.M., C.M.W., and T.A.S. designed the research. T.T., S.M., C.M.W., R.O., and T.A.S. supervised the project. P.N. and T.A.S. designed and made the GFP-integrin constructs. T.I.M. performed imaging experiments and analyzed data. P.N. wrote image processing and analysis code and analyzed data. S.B.M. and J.M.K. set up polarization microscopes and R.O. advised on analysis. N.K., T.A.S., and D.B. performed Rosetta modeling. T.A.S. constructed the integrin-microscope reference frame and defined the G.F.P. transition dipole orientation within Rosetta models. T.A.S. and S.B.M. analyzed Rosetta data. T.I.M., T.J.L., and J.C.W. performed SIM experiments and analysis. S.B.

M., J.M.K., and V.S. contributed to image data analysis. P.N., T.I.M., and T.A.S. drafted the manuscript. J.M.K. and V.S. contributed equally to this work. All authors discussed the results and commented on the manuscript.

## Additional information

**Competing interests:** The authors declare no competing financial interests.

