## [Peer Review File · Nature Communications]

Reviewers' comments:

Reviewer #1 (Remarks to the Author):

The manuscript by Nordenfelt et al investigates the order and orientation of alpha-beta integrin hetro-dimers. The use of fluorescence polarization microscopy, is powerful and elegant. The fluorescence controls, including a GFP with more flexibility tagged in the same location are appropriate and nice. This approach is uniquely suited to identify nano- or angstrom scale protein orientations within a cellular context; allowing investigation of hypothesis related to receptor orientation or organization. The work is novel, and the question of great interest. Questions are raised related to how the modeling is integrated with data interpretation and importantly, if the claim of the title is fully supported by the data as presented.

Major points

The measured whole cell anisotropy of GFP is significantly less than the reported anisotropy of GFP in cells from both microscopy and spectroscopy (approximately 0.32). Why is the anisotropy measured here so much lower?

In Figure 2, the orientation of the molecules relative to the excitation polarization is not considered. If the molecules are all oriented, but not parallel to the ex polarization, this will impact the anisotropy measurement. In fact this feature is used in Figure 4 to establish orientation relative to the leading edge of cells. This should be discussed/considered.

Actin flow relative to the membrane tangent while centered at 90 appears widely distributed (Fig 3b) compared to the distribution of integrin relative to membrane normal (Fig 5g). The claim that actin flow is orienting the integrin should be weakened in light of this discrepancy. In fact, the two distributions are not compared. Imaging of integrin orientation and actin flow in the same cells is needed to address the connection between actin flow and orientation of LFA-1.

The modeling was confusing and could be more clearly explained. Where is the plasma membrane in the microscope coordinate system? The bounds placed on the correlation between the experiment and the model need to be more clearly explained and justified, specifically in the second paragraph on page 9 beginning "Both integrin...". Measurements from a second GFP construct, where the GFP dipole is either at a different orientation relative to the beta propeller or in a second location all together, would greatly assist in orienting the integrin molecule.

Minor points

Page 5 middle paragraph "Measured anisotropy was consistent with the predicted GFP dipole alignment among members of structural ensembles obtained via Rosetta" Explain how this was determined, how many members of the ensemble are classified as consistent?

Reviewer #2 (Remarks to the Author):

The manuscript by Nordenfelt et al. shows that the direction of retrograde actin flow influences integrin LFA-1 orientation in leukocytes. The data seem convincing and support this particular finding. However, other interpretations about integrin structure and orientation presented in this manuscript are speculative.

While I agree with the authors' claim that integrin orientation aligns with the retrograde flow, it is unclear whether data enable further interpretation.

For example, the experiments with Mn^{2+} are inconclusive. They show less orientation and the authors claim that integrins uncouple from the actin cytoskeleton. However, there is no evidence for this. There maybe multiple other interpretations; for example, the retrograde flow may be perturbed because of outside-in signals, or integrins in high affinity may orientate along the orientation of ECM ligands rather than the retrograde flow.

The authors previously showed that increased integrin activity and increased forces on integrins decreases the retrograde flow. How does orientation correlate with adhesion strength and forces? Can integrins follow retrograde flow if they are inactive or in a primed state? How does it precisely follow the clutch mechanism idea? Would we not expect that blebbistatin has an effect on the retrograde flow if forces are involved?

I have difficulties to follow how the data "makes it inescapable to discuss integrin structure biology in the context of force application by the retrograde flow". Maybe this is the case, but at this stage I have problems following the conclusion that one can correlate orientation with structural integrin features. So far, the conclusions seem very speculative and deserve more experiments.

The authors have shown that talin head expression and Mn^{2+} treatment uncouples integrins from retrograde flow. Can one increase coupling to actin? Maybe through integrin activating mutants (and blocking mutants as controls)? The use of integrin activating antibodies (and blocking antibodies as control)?

To my opinion these points need clarification before publication.

We thank the reviewers for their positive and helpful comments and have the following responses:

Reviewer #1 (Remarks to the Author):

The manuscript by Nordenfelt et al investigates the order and orientation of alpha-beta integrin heterodimers. The use of fluorescence polarization microscopy, is powerful and elegant. The fluorescence controls, including a GFP with more flexibility tagged in the same location are appropriate and nice. This approach is uniquely suited to identify nano- or angstrom scale protein orientations within a cellular context; allowing investigation of hypothesis related to receptor orientation or organization. The work is novel, and the question of great interest. Questions are raised related to how the modeling is integrated with data interpretation and importantly, if the claim of the title is fully supported by the data as presented.

Thank you for these very positive comments.

Major points

The measured whole cell anisotropy of GFP is significantly less than the reported anisotropy of GFP in cells from both microscopy and spectroscopy (approximately 0.32). Why is the anisotropy measured here so much lower?

While many factors may be at play, the most salient is high-NA imaging in our EA-TIRFM setup. The magnitude of anisotropy of an ensemble ranges between 1 to -0.5 nominally. However, high-NA imaging introduces mixing of polarizations, reducing this range to 0.8 to -0.4 as shown in new Supplemental Fig. 1.

In response to this query, we have also added new text in several places in the MS. In Methods we add a new section "**System effects on anisotropy**", where we state that "Fluorescence anisotropy is dependent on the experimental set up as well as properties of the fluorophore such as: 1) angular diffusion coefficient of tethered fluorophores 2) degree of alignment to the polarization axis when fluorophores experience rotational constraint, and 3) dominant orientation of fluorophores when they are aligned. The high numerical aperture used in TIRF-based imaging introduces polarization mixing, reducing the range of anisotropy magnitude from 1 to -0.5 to 0.8 to -0.4 (Supplemental Fig. 1). For GFP in solution, these effects result in a drop of the observed anisotropy from the typical value of 0.32¹ to around 0.18-0.20^{2,3}. Since these depolarization effects are similar across the various experimental systems used in this manuscript, the relative changes in anisotropy and not the absolute values of anisotropy are significant."

We have revised the main text to address this issue in the second paragraph of Results: "The high NA objectives used in TIRF somewhat lower the absolute values of anisotropy² (Supplemental Fig. 1); therefore, we focus here on comparisons of the relative values of anisotropy." New Supplemental Fig. 1 shows a simulation of the effects of both dipole ensembles and our TIRF set up.

We address this issue again in the section "**Translation of dipole orientation to integrin orientation on the cell surface.**" where we state that "To determine GFP-LFA-1 fusion orientation on cell surfaces we utilized measurements of dipole orientation, and not absolute values of anisotropy or polarization factor, since the latter values can be effected by how uniformly the integrins mediating cell migration are aligned to one another and the presence of unaligned, unengaged integrins in the same regions. Absolute values of anisotropy and polarization factor will also be affected by variation in GFP orientation relative to the integrin, which is expected based on variation in orientation found in Rosetta ensembles (Fig. 1e)."

In Figure 2, the orientation of the molecules relative to the excitation polarization is not considered. If the molecules are all oriented, but not parallel to the ex polarization, this will impact the anisotropy measurement. In fact this feature is used in Figure 4 to establish orientation relative to the leading edge of cells. This should be discussed/considered.

We agree that we did not describe this well in the first draft. We have revised the relevant text and also Fig. 2 to show how dipole orientation affects anisotropy. In the second paragraph in Results we state: “Our measurements of fluorescence intensity are for the ensemble of LFA-1-GFP fusions within individual pixels (Fig. 2b). For constrained GFP-LFA-1 fusions that are aligned with one another and when they are aligned in a direction such that the transition dipole is parallel to the electric field of excitation, more emitted fluorescence will be recorded on the parallel than perpendicular camera (Fig. 2c), resulting in higher emission anisotropy than when the dipoles are more normal to excitation (Fig. 2b, compare pixels 2 and 3). In contrast, variation in GFP orientation relative to the integrin in unconstrained GFP (Fig. 2d) or random orientation of constrained GFP-LFA-1 fusions in the plane of the membrane (Fig. 2b, pixel 1) will lead to similar intensities recorded in the parallel and perpendicular cameras, and hence lower emission anisotropy.”

Actin flow relative to the membrane tangent while centered at 90 appears widely distributed (Fig 3b) compared to the distribution of integrin relative to membrane normal (Fig 5g). The claim that actin flow is orienting the integrin should be weakened in light of this discrepancy. In fact, the two distributions are not compared. Imaging of integrin orientation and actin flow in the same cells is needed to address the connection between actin flow and orientation of LFA-1.

The slightly wider distribution of actin flow angles compared to integrin orientation is most likely a result of the optical flow analysis of the SIM time-lapse images. In this analysis, little filtering or averaging was used to track the flow of actin, unlike in speckle microscopy⁴ where speckle intensity and flow direction are filtered. The optical flow analysis results in a noisier result where all of the actin filaments emerging from the membrane cortex are considered. Additionally, the distributions of actin flow direction and GFP-integrin orientation are not completely comparable as they differ in how the ROIs are averaged. In the actin flow analysis, the vectors of actin flow are averaged in the ROI, as shown in the enlarged images in Fig. 3a which approximately correspond to one ROI. In contrast, the GFP-LFA-1 dipole orientations are first measured as an ensemble of molecules in each pixel (average orientation / pixel) and then these ensembles are averaged in each ROI. This inherent additional averaging would be expected to result in a tighter distribution.

It would be ideal to measure actin flow and integrin orientation simultaneously within the migrating cell with live-cell actin flow measurements. We were not able to visualize and measure actin flow on ICAM-1 substrates with TIRF. We found that the use of live structured illumination microscopy (SIM) was required to visualize and measure actin flow in T-cells migrating on ICAM-1 substrates. SIM is not compatible with the polarization techniques needed to measure integrin orientation. Additionally, fluorescent speckle microscopy of actin flow can not currently be combined with anisotropy measurements due to the high frame rate and relatively high illumination conditions required to measure actin flow using this technique and GFP orientation using EA-TIRFM.

While accurate measurements of actin orientation are difficult, other measurements on T cell receptors and integrins on lymphocytes and integrin-engaged ICAM-1 in bilayers underneath lymphocytes have clearly shown that these actin-engaged molecules move centripetally, and we believe that it is reasonable to conclude that the distribution of actin flow velocities we see is a reflection of measurement error rather than a true reflection of heterogeneity in flow direction.

We have rewritten the MS to address these points. In Results, “**Quantitative description of actin flow in migrating T-cells**” we state: “We do not believe that the large apparent spread in actin flow orientation around the mean value of 90° in Fig. 3b is meaningful for several reasons. Measuring actin flow is always challenging; the alternative technique of speckle microscopy utilizes extensive denoising and averaging⁴; much less denoising is applied in optical flow analysis⁵. Furthermore, assigning the membrane tangent was often challenging as exemplified in the highly irregularly shaped cell on ICAM-1 in Fig. 3a. Finally, visual inspection confirmed the 90° value. In Fig. 3a, actin flow direction in the cells is color-coded; direction from the center of the inset circles is coded as the color at the perimeter of the circles. Thus despite irregularity in shape of the cell on ICAM-1 in Fig. 3a, in the orientation shown in the figure, flow at the top of cell is

largely towards the bottom (yellow), flow on the right of the cell is largely toward the left (red), and flow on the left of the cell is largely toward the right (green). Actin flow in migrating lymphocytes is thus centripetal, as previously found in lymphocyte immune synapses^{6,7}.

We address it again in Discussion: "We also made measurements of actin retrograde flow. Comparison of retrograde flow velocity on ICAM-1, CD43 antibody, and mixed ICAM-1/CD43 substrates demonstrated that LFA-1 slows actin retrograde flow in migrating T cells. The simplest interpretation of this result is that actin retrograde flow exerts a force on LFA-1. We found that flow was largely retrograde, i.e. normal to leading edge. We believe that experimental spread in our angular measurements of actin flow direction in migrating lymphocytes is due to noise, irregularity in cell shape, and the use of a method with less denoising than speckle microscopy. Measurements of actin flow with speckle microscopy in immunological synapses formed by lymphocytes have clearly demonstrated retrograde actin flow in a direction normal to the cell edge^{6,7}."

The modeling was confusing and could be more clearly explained. Where is the plasma membrane in the microscope coordinate system? The bounds placed on the correlation between the experiment and the model need to be more clearly explained and justified, specifically in the second paragraph on page 9 beginning "Both integrin...". Measurements from a second GFP construct, where the GFP dipole is either at a different orientation relative to the beta propeller or in a second location all together, would greatly assist in orienting the integrin molecule.

We have rewritten the MS to make the reference frame clearer. New panel a in Fig. 6 shows the membrane, cover slip, and objective, as well as the interacting molecules.

We have placed more bounds on correlation between experimental results and the retrograde flow model and added a caveat on the Rosetta modeling. We now fully address the results on α L-F and include additional panels in Fig. 6 to show α L-F.

New text is added to Results: "The experimentally determined value for the orientation of the transition dipole for α L-T is in perfect agreement with an orientation on the cell surface of LFA-1 molecules engaged to actin and ICAM-1 of $\theta = 0^\circ$. The experimentally determined dipole for α L-T of $\theta_d = 95.4 \pm 10.1^\circ$, shown in Fig. 6d as a solid black line with ± 1 s.d. shown with dashed black lines, aligns with the ensemble dipole predicted by Rosetta (green line) when orientation of the integrin in its reference frame is $\theta = 0^\circ$, over a wide range of ϕ tilts (Fig. 6d). This means that the integrin is oriented on the cell surface with its α -subunit - β -subunit axis parallel to the direction of actin flow as shown in Fig. 6b,c. This alignment is as predicted by the hypothesis that retrograde actin flow exerts a force on the β -subunit cytoplasmic domain that is resisted by the ligand. Note that transition dipoles have a 2-fold symmetry axis; the transition dipole orientation we measure is therefore compatible with an integrin orientation of either $\theta = 0^\circ$ or 180° . Nonetheless, evidence that the cytoskeleton applies the force to the integrin β -subunit forces us to choose β retrograde of α , which is fulfilled only by $\theta = 0^\circ$. Both integrin head rotation θ in the xy plane and tilt ϕ relative to the z axis will affect the measured projection of the GFP transition dipole in the xy plane (Fig. 6b,c). The effect of tilting LFA-1 is shown by comparison of Fig. 6b ($\theta = 0^\circ$, $\phi = 0^\circ$) with Fig. 6c ($\theta = 0^\circ$, $\phi = 45^\circ$). Tilting changes the orientation of the transition dipole in 3-dimensional space (red double-ended arrow) and its projection on the xy plane which we measure experimentally (orange double ended arrow) (Fig. 6b,c). Exploring a range of tilts of α L-T (variation in ϕ) shows that the transition dipole calculated from Rosetta ensemble members (green line in Fig. 6d) falls within one standard deviation of the experimentally determined θ_d value (black line in Fig. 6d) for ϕ values between 67.5° and 22.5° (Fig. 6d and Supplemental Fig. 10). The measured θ_d value of $71.7^\circ \pm 17.9^\circ$ for α L-F (solid black line, Fig. 6e) is also in perfect agreement with the calculated Rosetta ensemble transition dipole (Fig. 6e, black line); the calculated transition dipole falls within one standard deviation of the measured dipole for $\theta = 0^\circ$ for ϕ values between 45° and 11.25° (Fig. 6e and Supplemental Fig. 9). Considering that our transition dipole measurement for α L-T has an uncertainty of $\pm 10.1^\circ$, our results define the orientation of engaged LFA-1 on cell surfaces with respect to the reference frame as within about 10° of $\theta = 0$ and within about 25° of $\phi = 45$. A

caveat is that the average orientation that the GFP and its dipole adopt biologically with respect to the integrin may deviate from that predicted from Rosetta ensembles. However, α L-T deletes 5 N-terminal residues and 1 C-terminal residue from GFP to minimize both flexibility and uncertainty in orientation (Fig 1d). The range of transition dipole orientations found by Rosetta is relatively narrow for α L-T (Fig. 6d) and we expect that the biological orientation is included within the range of orientations sampled by Rosetta, and not far from the predicted orientation. Our confidence in our biological result and methodology is enhanced by the independent results with α L-F. It deletes no GFP residues and has a different predicted Rosetta ensemble dipole orientation and an experimentally measured θ_d value that differs by 24° from α L-T, yet also shows that LFA-1 has an orientation in the reference frame at close to $\theta=0$, in alignment with actin retrograde flow.”

We state in Discussion first paragraph: “We demonstrate that the GFP dipoles of GFP-LFA-1 fusions, and hence the LFA-1 moiety of these fusions, adopts a specific orientation with respect to the adjacent leading edge. Furthermore, with two GFP-LFA-1 fusions with distinct predicted orientations between the GFP dipole and the integrin, we define the orientation of the integrin with respect to the leading edge. This orientation is with an uncertainty of 10° exactly that predicted by the cytoskeletal force model of integrin activation, with the axis of tensile force transmission between the β -subunit cytoplasmic domain and the ligand ICAM-1 oriented in the same direction as actin retrograde flow. We obtained similar results on anisotropy / polarization factor, alignment, and orientation with two distinct fluorescent polarization microscope techniques, EA-TIRFM and Instantaneous FluoPolScope.”

We state in Discussion second paragraph: “Our experimental results suggest that tensile force constrains integrin tilt. The force vector applied by actin retrograde flow may be deconstructed into components parallel and normal to the membrane and it is to be expected that the parallel component is larger. Indeed, super-resolution measurements on talin show that it tilts in focal adhesions at an angle of $\sim 75^\circ$ with respect to the membrane normal⁸. We measured here a tilt (ϕ angle) of $\sim 45 \pm 25^\circ$ for LFA-1. Orientation in the xy plane (θ) is much better determined for α L-T than tilt because the orientation of the dipole in three dimensions is close to the xy plane; thus, ϕ has much less effect than θ on the experimental measurable of the projection on the xy plane of the dipole (compare red dipole and its gold projection at $\phi = 0$ and 45° in Fig. 6b and 6c, respectively).”

Minor points

Page 5 middle paragraph “Measured anisotropy was consistent with the predicted GFP dipole alignment among members of structural ensembles obtained via Rosetta” Explain how this was determined, how many members of the ensemble are classified as consistent?

We have revised this section and added Supplemental Table 2 to show polarization factors. This section now states: “Emission anisotropy was highest for the truncated construct, α L-T; intermediate for the full length construct, α L-F; and lowest for the construct with added linker residues, α L-L, which had an emission anisotropy comparable to cytoplasmic GFP (Fig. 2e-f). These findings were in agreement with Rosetta results, which showed that α L-T ensemble members had well aligned transition dipoles (Fig. 1e) and the highest calculated polarization factor (Supplemental Table 2).”

Reviewer #2 (Remarks to the Author):

The manuscript by Nordenfelt et al. shows that the direction of retrograde actin flow influences integrin LFA-1 orientation in leukocytes. The data seem convincing and support this particular finding. However, other interpretations about integrin structure and orientation presented in this manuscript are speculative.

Thank you for your comments.

While I agree with the authors' claim that integrin orientation aligns with the retrograde flow, it is unclear whether data enable further interpretation.

For example, the experiments with Mn²⁺ are inconclusive. They show less orientation and the authors claim that integrins uncouple from the actin cytoskeleton. However, there is no evidence for this. There maybe multiple other interpretations; for example, the retrograde flow may be perturbed because of outside-in signals, or integrins in high affinity may orientate along the orientation of ECM ligands rather than the retrograde flow.

Uncoupling was a bad choice of words. Furthermore, how Mn works and its effects on alignment are very minor points in our MS and we have decreased the discussion of Mn in the revised MS. We have revised our comments on its effects on retrograde flow in the section on actin flow: "Activating LFA-1 with Mn²⁺ increased actin flow velocity (28 ± 2 nm/sec) relative to untreated cells (Supplemental Fig. 3), suggesting that artificial activation with Mn²⁺ bypasses the need for integrin association with actin for integrin activation, and leads to a decrease in LFA-1 association with the cytoskeleton."

The authors previously showed that increased integrin activity and increased forces on integrins decreases the retrograde flow. How does orientation correlate with adhesion strength and forces? Can integrins follow retrograde flow if they are inactive or in a primed state? How does it precisely follow the clutch mechanism idea? Would we not expect that blebbistatin has an effect on the retrograde flow if forces are involved?

We believe the reviewer is referring to a different kind of finding in the Nordenfelt paper. When we mutated the binding sites for talin and kindlin in the integrin, actin retrograde flow velocity increased and force on integrins was abolished. Therefore, we found no quantitative relationship between flow velocity and force on the integrin, just a relationship between velocity and absence or presence of force. I would expect that the degree of integrin alignment and orientation would be similar over a wide range of forces, just as the chain of a necklace would have a similar alignment and orientation over a wide range of forces.

We think the clutch mechanism was used imprecisely and we have dropped it.

Blebbistatin is a myosin inhibitor; acto-myosin contraction appears to have little role in lymphocytes. The major driver of retrograde flow at the leading edge is actin polymerization at the membrane pushing the actin filamentous actin backwards into the cell. We have revised Results to discuss these concepts: "Blebbistatin, an inhibitor of myosin-dependent actin filament contractility, had no effect on α L-T anisotropy (Fig. 3d-e), consistent with the lack in T cells of actin stress fibers. In contrast, cytochalasin D, an inhibitor of actin polymerization, significantly decreased LFA-1 anisotropy (Fig. 3d-e). Together, these results show that an intact cytoskeleton is required for LFA-1 anisotropy and suggest that actin polymerization, a mechanism operative in the lamellipodium to generate retrograde flow that is independent of actomyosin contraction⁹, is important for LFA-1 anisotropy and hence alignment."

Our recent thermodynamic work suggests that only integrins that are engaged intracellularly to a force-generating cytoskeleton and extracellularly to a ligand will be aligned. The reviewer asks a question that is likely to be on many reader's minds and we have addressed it by adding a new paragraph to the Discussion: "Our fluorescence measurements here are for the ensemble of all three conformational states of our GFP-LFA-1 fusions. Based on thermodynamic measurements and calculations, essentially all integrins that are simultaneously bound to a force-generating cytoskeleton and to ICAM-1 are in the extended-open conformation, whereas integrins that are unbound or bound only to adaptors or ICAM-1 are predominantly in the bent-closed conformation¹⁰. Only the adaptor and ligand-engaged, extended-open integrins can be aligned by actin retrograde flow and contribute to the alignment measured as anisotropy and polarization factor and the orientation measured as dipole orientation here. Unengaged, bent-closed integrins will be randomly oriented on the cell surface, are likely to make up a substantial and perhaps the majority of the integrin ensemble on the cell surface both toward the center of cells and at the leading edge, and contribute to lowering observed anisotropy and polarization factor values."

I have difficulties to follow how the data “makes it inescapable to discuss integrin structure biology in the context of force application by the retrograde flow”. Maybe this is the case, but at this stage I have problems following the conclusion that one can correlate orientation with structural integrin features. So far, the conclusions seem very speculative and deserve more experiments.

We were trying to emphasize the importance of considering structures in the light of force; “inescapable” was perhaps inappropriate and we have rewritten this section. We do not think this section is speculative; everything follows from simple physical principles such as force balance. We have now followed this paragraph with a second that discusses recent advances in understanding integrins from measuring their thermodynamics. These two paragraphs in Discussion read:

“In contrast to force measurements in cells, crystal, EM, NMR, SAXS, and neutron scattering structures of integrins are determined in the absence of force. Elegant structures have revealed intact integrins, integrin ectodomains and their complexes with ligands, integrin transmembrane domains, and integrin cytoplasmic domains and their complexes with intracellular effectors that link or inhibit linkage to the cytoskeleton¹¹⁻¹⁶. The demonstration here that we can use fluorescence microscopy to define a specific orientation for integrin atomic structures on the surface of migrating cells now places these structures not only in the context of intact functioning cells, but makes it essential to discuss integrin structural biology in the context of force application by actin retrograde flow. For example, flexibility between many of the domains in integrins enables them to straighten and align with their domain-domain junctions parallel to the force. Flexibility in poorly structured residues that link the ectodomain to the plasma membrane enables integrin tilting¹⁷ as suggested by our measurements here. Moreover, the integrin β -subunit transmembrane domain tilts when separated from the α -subunit transmembrane domain, as occurs upon integrin activation¹¹⁻¹³, consistent with the tilt suggested here.”

“While almost every paper on an integrin complex discusses how binding of a ligand or effector may regulate integrin activation by selecting among integrin conformational states, the implications of complex formation for tensile force transmission from the cytoskeleton that could select among integrin conformational states is discussed by few workers in the field. Our findings on integrin orientation on cell surfaces provide strong support for the cytoskeletal force model of integrin activation^{11,18-22}. Recent thermodynamic measurements and calculations provide further insights by showing that force enables ultrasensitive regulation of integrin adhesiveness^{10,23}. Among the three integrin conformational states shown in Fig. 1a, only the extended-open state has high affinity for ligand^{11,23}. This affinity has been measured as ~1,000-fold higher for integrin LFA-1²¹ and more precisely as 4,000-fold higher for integrin $\alpha 5 \beta 1$ for the extended-open than the bent-closed and extended-closed conformations²³. Measurements of free energies of integrin conformational states on the cell surface have shown that the bent-closed conformation is substantially lower in free energy than the extended-closed or extended-open conformation, with >99% of integrins in the bent-closed conformation at equilibrium²³. Binding of adaptors such as talin has little effect on this equilibrium¹⁰. However, tensile force stabilizes equilibria by a potential energy equal to the force times the difference between each state in extension along the force bearing axis. Thermodynamic calculations show that in an equilibrium model in which integrins are allowed to bind intracellular adaptors and extracellular ligands, and tensile force is applied when both adaptor and ligand bind, the extended-open conformation is greatly stabilized, and becomes the predominant state. Force in the same range as measured on integrins and ligands, in the 2 pN range, is sufficient to fully activate integrins. Moreover, whereas adaptors such as talin have little effect in the absence of force, they provide ultrasensitive integrin activation in the presence of force¹⁰.”

The authors have shown that talin head expression and Mn2+ treatment uncouples integrins from retrograde flow. Can one increase coupling to actin? Maybe through integrin activating mutants (and blocking mutants as controls)? The use of integrin activating antibodies (and blocking antibodies as control)?

This MS adds to the evidence that force application by the actin cytoskeleton is the physiologic mechanism for activating integrins. Activating integrins by mutations or antibodies would be similar to Mn in bypassing the need for activation by the cytoskeleton. Therefore, we would expect such manipulations, like Mn, to increase the rate of actin retrograde flow. However, this would go beyond the main point of the paper, which is to provide evidence that the actin cytoskeleton aligns and orients integrins. We believe that our story is quite complete and takes a big step forward, and that many of the types of questions the reviewer is interested in are well addressed by our more elaborate Discussion, which now includes the two paragraphs cited above.

To my opinion these points need clarification before publication.

In addition to the changes prompted by the reviewers comments, we have also carefully revised the text again to improve it. We have included a supplemental file with tracked changes to show all changes.

References

- 1 Rocheleau, J. V., Edidin, M. & Piston, D. W. Intrasequence GFP in class I MHC molecules, a rigid probe for fluorescence anisotropy measurements of the membrane environment. *Biophys. J.* **84**, 4078-4086, doi:10.1016/S0006-3495(03)75133-9 (2003).
- 2 Ghosh, S., Saha, S., Goswami, D., Bilgrami, S. & Mayor, S. Dynamic imaging of homo-FRET in live cells by fluorescence anisotropy microscopy. *Methods Enzymol.* **505**, 291-327, doi:10.1016/B978-0-12-388448-0.00024-3 (2012).
- 3 Axelrod, D. Fluorescence microscopy of living cells in culture. Part B. Quantitative fluorescence microscopy--imaging and spectroscopy. *Methods Cell Biol.* **30**, 1-498 (1989).
- 4 Danuser, G. & Waterman-Storer, C. M. Quantitative fluorescent speckle microscopy of cytoskeleton dynamics. *Annu. Rev. Biophys. Biomol. Struct.* **35**, 361-387, doi:10.1146/annurev.biophys.35.040405.102114 (2006).
- 5 Lucas, B. D. & Kanade, T. in *Proceedings of the 7th international joint conference on Artificial intelligence - Volume 2* 674-679 (Morgan Kaufmann Publishers Inc., Vancouver, BC, Canada, 1981).
- 6 Kaizuka, Y., Douglass, A. D., Varma, R., Dustin, M. L. & Vale, R. D. Mechanisms for segregating T cell receptor and adhesion molecules during immunological synapse formation in Jurkat T cells. *Proc. Natl. Acad. Sci. U S A* **104**, 20296-20301 (2007).
- 7 Comrie, W. A. & Burkhardt, J. K. Action and Traction: Cytoskeletal Control of Receptor Triggering at the Immunological Synapse. *Front Immunol.* **7** (2016).
- 8 Liu, J. *et al.* Talin determines the nanoscale architecture of focal adhesions. *Proc. Natl. Acad. Sci. U. S. A.* **112**, E4864-4873, doi:10.1073/pnas.1512025112 (2015).
- 9 Ponti, A., Machacek, M., Gupton, S. L., Waterman-Storer, C. M. & Danuser, G. Two distinct actin networks drive the protrusion of migrating cells. *Science* **305**, 1782-1786, doi:10.1126/science.1100533305/5691/1782 [pii] (2004).
- 10 Li, J. & Springer, T. A. Integrin extension enables ultrasensitive regulation by cytoskeletal force. *Proc. Natl. Acad. Sci. U. S. A.*, doi:10.1073/pnas.1704171114 (2017).

- 11 Springer, T. A. & Dustin, M. L. Integrin inside-out signaling and the immunological synapse. *Curr. Opin. Cell Biol.* **24**, 107-115, doi:10.1016/j.ceb.2011.10.004 (2012).
- 12 Kim, C., Ye, F. & Ginsberg, M. H. Regulation of integrin activation. *Annu. Rev. Cell Dev. Biol.* **27**, 321-345, doi:10.1146/annurev-cellbio-100109-104104 (2011).
- 13 Campbell, I. D. & Humphries, M. J. Integrin structure, activation, and interactions. *Cold Spring Harb. Perspect. Biol.* **3**, doi:10.1101/cshperspect.a004994 (2011).
- 14 Calderwood, D. A., Campbell, I. D. & Critchley, D. R. Talins and kindlins: partners in integrin-mediated adhesion. *Nat. Rev. Mol. Cell Biol.* **14**, 503-517, doi:10.1038/nrm3624 (2013).
- 15 Liu, J. *et al.* Structural mechanism of integrin inactivation by filamin. *Nat. Struct. Mol. Biol.* **22**, 383-389, doi:10.1038/nsmb.2999 (2015).
- 16 Bouvard, D., Pouwels, J., De Franceschi, N. & Ivaska, J. Integrin inactivators: balancing cellular functions in vitro and in vivo. *Nat. Rev. Mol. Cell Biol.* **14**, 430-442, doi:10.1038/nrm3599 (2013).
- 17 Zhu, J. *et al.* The structure of a receptor with two associating transmembrane domains on the cell surface: integrin $\alpha_{IIb}\beta_3$. *Mol. Cell* **34**, 234-249 (2009).
- 18 Astrof, N. S., Salas, A., Shimaoka, M., Chen, J. F. & Springer, T. A. Importance of force linkage in mechanochemistry of adhesion receptors. *Biochemistry* **45**, 15020-15028 (2006).
- 19 Alon, R. & Dustin, M. L. Force as a Facilitator of Integrin Conformational Changes during Leukocyte Arrest on Blood Vessels and Antigen-Presenting Cells. *Immunity* **26**, 17-27 (2007).
- 20 Zhu, J. *et al.* Structure of a complete integrin ectodomain in a physiologic resting state and activation and deactivation by applied forces. *Mol. Cell* **32**, 849-861 (2008).
- 21 Schürpf, T. & Springer, T. A. Regulation of integrin affinity on cell surfaces. *EMBO J.* **30**, 4712-4727, doi:doi: 10.1038/emboj.2011.333 (2011).
- 22 Nordenfelt, P., Elliott, H. L. & Springer, T. A. Coordinated integrin activation by actin-dependent force during T-cell migration. *Nature communications* **7**, doi:10.1038/ncomms13119 (2016).
- 23 Li, J. *et al.* Conformational equilibria and intrinsic affinities define integrin activation. *EMBO J.* **36**, 629, doi:10.15252/embj.201695803 (2017).

REVIEWERS' COMMENTS:

Reviewer #1 (Remarks to the Author):

The authors have satisfied my concerns with this manuscript. I feel this work is novel and of great interest.